# Median raphe serotonergic neurons projecting to the interpeduncular nucleus control preference and aversion

Hiroyuki Kawai[1,2,6], Youcef Bouchekioua [3,6], Naoya Nishitani[1,3,4], Kazuhei Niitani[4], Shoma Izumi[4], Hinako Morishita[1], Chihiro Andoh[1], Yuma Nagai[1], Masashi Koda[1], Masako Hagiwara[1], Koji Toda[5], Hisashi Shirakawa [1], Kazuki Nagayasu [1]✉, Yu Ohmura [3]✉, Makoto Kondo[2], Katsuyuki Kaneda [4], Mitsuhiro Yoshioka[3] & Shuji Kaneko [1]✉

Appropriate processing of reward and aversive information is essential for survival. Although a critical role of serotonergic neurons in the dorsal raphe nucleus (DRN) in reward processing has been shown, the lack of rewarding effects with selective serotonin reuptake inhibitors (SSRIs) implies the presence of a discrete serotonergic system playing an opposite role to the DRN in the processing of reward and aversive stimuli. Here, we demonstrated that serotonergic neurons in the median raphe nucleus (MRN) of mice process reward and aversive information in opposite directions to DRN serotonergic neurons. We further identified MRN serotonergic neurons, including those projecting to the interpeduncular nucleus (5-HT$^{MRN \to IPN}$), as a key mediator of reward and aversive stimuli. Moreover, 5-HT receptors, including 5-HT$_{2A}$ receptors in the interpeduncular nucleus, are involved in the aversive properties of MRN serotonergic neural activity. Our findings revealed an essential function of MRN serotonergic neurons, including 5-HT$^{MRN \to IPN}$, in the processing of reward and aversive stimuli.

Processing reward and aversive information is essential for the survival of living organisms[1–10]. Aberrant neural activity underlying this processing can cause under- and overestimation of the value of extrinsic stimuli, which are frequently seen in patients with mental disorders including drug addiction and major depression[11–13]. Therefore, it is of high importance to better understand the neuronal mechanisms underlying the processing of reward and aversion. Previous reports have revealed the importance of serotonin in this process. Although the dorsal raphe nucleus (DRN) and median raphe nucleus (MRN) are the main origins of serotonergic

projections to the forebrain, most studies have focused on the DRN and revealed that optogenetic activation of serotonergic neurons in the DRN is rewarding[14,15] while DRN serotonergic neurons are activated in response to reward stimuli as well as aversive stimuli[16–21]. However, the systemic administration of selective serotonin reuptake inhibitors (SSRIs) does not exert clear rewarding effects[22–24] or satisfactory therapeutic effects for anhedonia[25], a main symptom of depression, although SSRIs increase extracellular levels of serotonin in regions that the DRN and MRN project to[26–28]. These contradictory findings imply that serotonin neurons in the MRN

[1]Department of Molecular Pharmacology, Graduate School of Pharmaceutical Sciences, Kyoto University, 46-29 Yoshida-Shimoadachi-cho, Sakyo-ku, Kyoto 606-8501, Japan. [2]Department of Anatomy and Neuroscience, Graduate School of Medicine, Osaka Metropolitan University, 1-4-3 Asahi-cho, Abeno-ku, Osaka 545-8585, Japan. [3]Department of Neuropharmacology, Faculty of Medicine and Graduate School of Medicine, Hokkaido University, N15 W7 Kita-ku, Sapporo 060-8638, Japan. [4]Laboratory of Molecular Pharmacology, Institute of Medical, Pharmaceutical and Health Sciences, Kanazawa University, Kanazawa 920-1192, Japan. [5]Department of Psychology, Keio University, 2-15-45 Mita, Minato-ku, Tokyo 108-8345, Japan. [6]These authors contributed equally: Hiroyuki Kawai, Youcef Bouchekioua. ✉e-mail: nagayasu@pharm.kyoto-u.ac.jp; yohmura@med.hokudai.ac.jp; skaneko@pharm.kyoto-u.ac.jp

might play an opposite role to the DRN in the processing of reward and aversive stimuli.

Some research has suggested a critical role of the MRN in the processing of reward and aversive stimuli. Although not highly selective to serotonin neurons, pharmacological inactivation of the MRN induced conditioned place preference and ethanol reinstatement[29,30], while electrical stimulation of the MRN suppresses lever pressing associated with reward[31]. More recently, cell-type selective recordings and manipulation showed that aversive stimuli increased the activity of vesicular glutamate transporter 2 (vGluT2)-expressing MRN glutamatergic neurons and that activation of these neurons induced place avoidance, although the role of serotonergic neurons in the MRN, which has a distinct projection pattern to vGluT2-expressing MRN glutamatergic neurons, has yet to be determined[32]. To clarify the precise role of MRN serotonin neurons in the processing of reward and aversive stimuli, we selectively recorded changes in serotonergic activity in the MRN and projection areas in response to rewarding and aversive stimuli. Next, we manipulated these serotonergic activities to determine whether their inhibition and activation were sufficient for mimicking behaviors induced by rewarding and aversive stimuli, respectively. Our findings demonstrate the opposite role of the MRN to the DRN in regulating the balance between preference and aversion, and further elucidate the serotonergic pathway and related 5-HT receptor subtypes.

## Results

### Appetitive reward stimuli inhibit MRN serotonergic neurons, and inhibition of MRN serotonergic neurons is rewarding

To measure changes in neuronal activity of MRN serotonergic neurons, an adeno-associated virus (AAV) bearing GCaMP6s[33] or Venus[34] control under the control of the mouse tryptophan hydroxylase 2 (Tph2) promoter was injected into the MRN[15,35]. Histological analyses of Venus-expressing animals revealed the high specificity and coverage of the promoter (specificity: $91.4 \pm 2.5\%$; coverage: $95.4 \pm 1.4\%$; $n = 5$; Fig. 1a–d). Then, we recorded MRN serotonergic neuron activity before and after sucrose consumption through fiber photometry in freely moving mice (Fig. 1e–i, Supplementary Fig. 1a–d, g). Individual analysis revealed that the GCaMP fluorescence started to decrease just after sucrose licking and that the fluorescence remained low throughout the bouts. These phase-dependent fluorescence changes were observed over almost every trial (Fig. 1g, h). The mean GCaMP fluorescence started to decrease after sucrose licking in the MRN of all mice (Fig. 1i). The average signal peak ($\Delta F/F_0$) for GCaMP mice and Venus control mice was $-2.51 \pm 0.78\%$ and $-0.31 \pm 0.45\%$, respectively (mean ± s.e.m., $n = 7$ (GCaMP) and $n = 6$ (Venus) mice; $t_{11} = 2.317$, *$P = 0.0408$; Supplementary Fig. 1e). To determine the relationship between the objective value of rewards and changes in serotonergic activity in the MRN, we compared GCaMP fluorescence changes after consumption of 10% and 0.5% sucrose solutions. We found that the size of fluorescence changes after the consumption of 0.5% sucrose was significantly smaller than that after the consumption of 10% sucrose (Fig. 1j). When we analyzed the GCaMP fluorescence changes in the same baseline setting as the previous report on DRN serotonergic neurons[16], we found a significant increase in GCaMP fluorescence from −1 to −0.5 s as well as a decrease after sucrose licking (Supplementary Fig. 1i–l). As the tested mice should stop locomotion to lick sucrose solution from the spout, licking onset was time-locked to the termination of locomotion. To determine the extent to which changes in GCaMP fluorescence were affected by the termination of locomotion, we analyzed GCaMP fluorescence in the MRN before and after the termination of locomotion and compared the extent of the changes to that of sucrose-induced changes. We found a small but significant GCaMP fluorescence decrease from 0.3 to 1.2 s after the termination of locomotion (Supplementary Fig. 1m), whereas the size of the signal changes (mean $\Delta F/F_0$) after the termination of locomotion was much smaller than that after sucrose licking ($t_6 = 2.935$, *$P = 0.0261$; Supplementary Fig. 1n).

To assess the causal relationship between decreases in MRN activity and appetitive characteristics, we tested the effect of optogenetic inhibition of MRN serotonin neurons in the conditioned place preference (CPP) test[15,36]. MRN serotonin neurons were selectively transduced with archaerhodopsin (eArchT), a light-activated proton pump, for optogenetic inhibition[37] (Fig. 1k, l and Supplementary Fig. 1h). Ex vivo electrophysiology experiment revealed that activity of eArchT-expressing cells was significantly attenuated by green light illumination and was reversed to the basal level after light illumination (before: $2.0 \pm 0.33$ Hz, light: $0.038 \pm 0.0028$ Hz, after: $1.5 \pm 0.33$ Hz, 30 sweeps in 6 cells from 3 mice; ***$P = 0.0005$ by one-way repeated measures ANOVA, **$P = 0.0062$ (before vs light), $P = 0.2209$ (before vs after), *$P = 0.0221$ (light vs after) by paired $t$-test, Supplementary Fig. 1o, p). In the pretest session, the mice were allowed to freely explore two chambers with different wall colors and textures to facilitate discrimination. The time spent in each chamber was recorded. In the conditioning session, the mice were confined to one of two chambers and allowed to explore without optogenetic stimulation for 20 min. After at least 4 h, the mice were confined to the other chamber for 20 min, during which they were subjected to optogenetic inhibition of MRN serotonergic neurons (cycles of 20-s light ON and 10-s light OFF) and received the same treatment on the following day. The posttest session was performed the day following the second conditioning session. In the posttest session, the mice were allowed to freely explore the two chambers again, and the time spent in each chamber was recorded. The CPP score, an index of the rewarding property of manipulation, was defined as the difference in the time spent in the optogenetic manipulation-paired chamber during the posttest and pretest sessions. The CPP scores of the eArchT-expressing animals were significantly higher than those of the Venus control animals (Fig. 1m). The increase in time spent in the optogenetic inhibition-paired chamber cannot be accounted for by locomotor suppression, as locomotor activity in the posttest session in the eArchT-expressing mice was slightly but significantly higher than that in Venus control mice (Supplementary Fig. 1f). Consistent with this finding, the time spent in the optogenetic inhibition-paired chamber significantly increased after conditioning sessions in the eArchT mice but not in the Venus mice (Fig. 1n). Optogenetic inhibition of 20 s per cycle in the CPP test was far longer than the duration of the decreased activity observed in our fiber photometry recordings (Fig. 1g–i). Therefore, we further investigated whether brief inhibition of MRN serotonergic neurons was sufficient for inducing reward-related behavioral changes in an operant conditioning task. In this task, optogenetic inhibition was briefly applied for 3 s when the mice performed nose poking (Fig. 1o). The number of nose pokes during the 30-min session increased in a time-dependent manner in the eArchT-expressing mice (Fig. 1p). These results indicated that inhibition of MRN serotonergic neurons reinforced reward-seeking behavior.

### Aversive stimuli activate MRN serotonergic neurons, and activation of MRN serotonergic neurons is aversive

To investigate the effects of an aversive stimulus on the neural activity of MRN serotonergic neurons, we measured GCaMP fluorescence in the MRN before and after a tail pinch (Fig. 2a, b and Supplementary Fig. 2h). The fluorescence in the MRN increased after a tail pinch in all mice (Fig. 2c), whereas it did not significantly change after control treatment (the mouse was approached with the clamp, but the tail was not pinched) (Supplementary Fig. 2d–f). The average signal peak ($\Delta F/F_0$) for the GCaMP mice and Venus control mice was $3.36 \pm 0.68\%$ and $0.04 \pm 0.54\%$, respectively (mean ± s.e.m., $n = 7$ mice; $t_{12} = 3.851$, **$P = 0.0023$; Supplementary Fig. 2a–c). As tail pinch increased locomotion, the tail pinch was time-locked to the initiation of locomotion. To determine the extent to which changes in GCaMP fluorescence were affected by the initiation of locomotion, we analyzed GCaMP fluorescence in the MRN before and after the initiation of locomotion

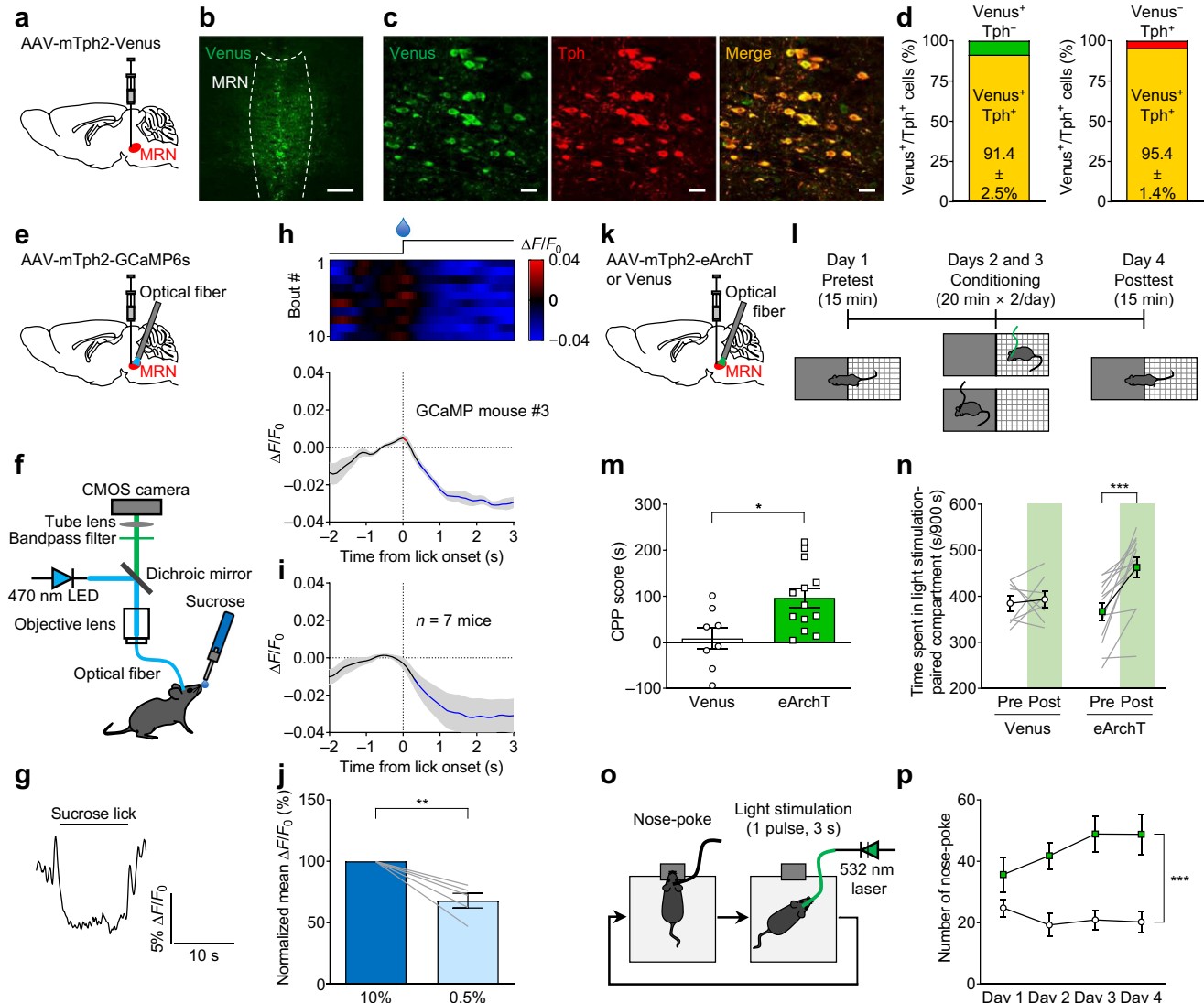

**Fig. 1 | MRN serotonergic neurons are inhibited by rewarding stimuli, and their inhibition elicits conditioned place preference (CPP) and self-stimulation.**
**a, e, k** Schematic representation of experiments. **b** AAV-mTph2-Venus injection into the MRN. Scale bar = 200 μm. **c, d** Colocalization rate of the transgene (Venus; green) and Tph2 (red) immunoreactivity in the MRN. Scale bars = 20 μm. $n = 5$ mice. **f** Fiber photometry setup. **g** Representative raw trace of GCaMP fluorescence. **h** Top: heatmap of signals (red–blue, high–low). One licking bout per row. Bottom: averaged GCaMP signals of one mouse. $n = 10$ bouts. **i** Mean GCaMP signals for seven mice. Lines and shaded areas indicate mean and s.e.m., respectively. Red and blue segments indicate a statistically significant increase and decrease from the baseline ($P < 0.05$; permutation test). **j** GCaMP fluorescence response to 10% and 0.5% sucrose solutions. Two-tailed paired $t$-test $t_4 = 5.365$, **$P = 0.0058$, $n = 5$ mice. **l** CPP test. **m** CPP scores calculated as spent time in a light stimulation-associated compartment in the posttest subtracted by that in the pretest in the Venus and

eArchT mice (two-tailed unpaired $t$-test, $t_{19} = 2.804$, **$P = 0.0113$, $n = 8$ (Venus) and 13 (eArchT) mice). **n** Spent time in the compartment associated with light stimulation was significantly increased after conditioning sessions in the eArchT mice but not in the Venus control mice (eArchT: two-tailed paired $t$-test (eArchT Pre vs eArchT Post), $t_{12} = 4.810$, ***$P = 0.0004$, $n = 13$ mice; Venus: two-tailed paired $t$-test (Venus Pre vs Venus Post), $t_7 = 0.3553$, $P = 0.7328$, $n = 8$ mice). **o** Schematic of the nose poke self-stimulation test. Greenlight (3 s duration, 1 pulse/poke, 5 mW) was delivered when mice performed nose poke responses. **p** The number of nose poke in the Venus and eArchT mice (two-way repeated-measures ANOVA, virus (Venus or eArchT) × time (day 1, 2, 3, or 4) interaction $F_{3,60} = 1.925$, $P = 0.1351$, the effect of virus $F_{1,20} = 38.40$, ***$P < 0.001$, the effect of time $F_{3,60} = 0.7157$, $P = 0.5464$, $n = 12$ (Venus) and 10 (eArchT) mice). Data are presented as mean ± s.e.m. Error bars indicate s.e.m. Source data are provided as a Source Data file.

and compared the size of the changes to that of tail pinch-induced changes. We found a small but significant GCaMP fluorescence increase after the initiation of locomotion (Supplementary Fig. 3a), whereas the size of the signal changes (mean $\Delta F/F_0$) after the initiation of locomotion was much smaller than that after tail pinch ($t_6 = 4.127$, **$P = 0.0062$; Supplementary Fig. 3b). In addition, we measured GCaMP fluorescence before and after licking with water and quinine solution, another representative aversive stimulus. As the tested mice were water-deprived, water itself was predicted to have a positive value. We observed a significant decrease in GCaMP fluorescence in response to water, whereas quinine (5 mM) blunted this decrease

(water: $t_{13} = 2.352$, *$P = 0.0351$; quinine: $t_{13} = 0.2024$, $P = 0.8428$; Supplementary Fig. 3c–f). We then measured GCaMP fluorescence in MRN serotonin neurons in response to auditory cues associated with foot shocks, using a cue-induced fear conditioning paradigm[38]. One day after training, GCaMP fluorescence in the MRN was measured before and after the onset of the auditory cues associated with foot shocks. We found that GCaMP fluorescence increased after the onset of the auditory cue ($P < 0.05$, $n = 5$ mice; Supplementary Fig. 3g).

To examine the causal relationship between this increase in activity and aversion-related behaviors, we expressed CheRiff, an excitatory optogenetic actuator[39], in MRN serotonergic neurons and

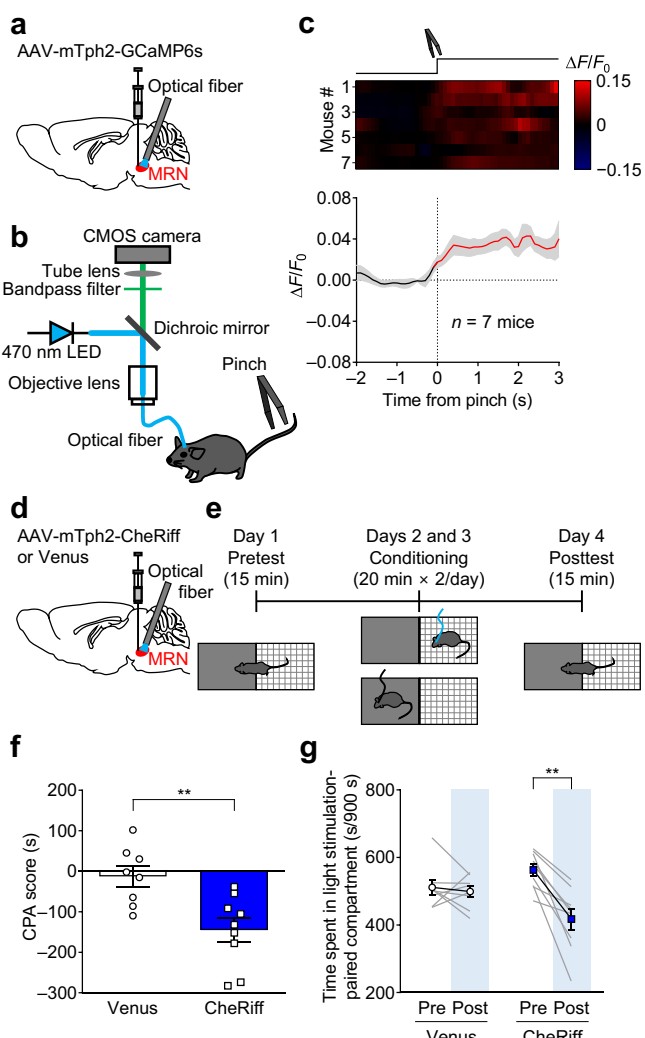

**Fig. 2 | Aversive stimuli activate MRN serotonergic neurons, and their activation elicits conditioned place aversion (CPA). a** Schematic representation of AAV injection and fiber implantation. **b** Schematic of the fiber photometry setup and tail pinch task in free-moving mice. **c** Top: heatmap of GCaMP signals (red–blue, high–low) for each trial (one trial per mouse). Each row represents the data of one mouse. $n = 7$ mice. Bottom: averaged traces of GCaMP signals. Lines and shaded areas indicate mean and s.e.m., respectively. The red segment indicates a statistically significant increase from the baseline ($P < 0.05$; permutation test).
**d** Schematic representation of AAV injection and fiber implantation. **e** Schematic of the CPA test. On day 1, mice underwent the two-compartment test without light stimulation (pretest: 15 min). On day 2 and 3, mice underwent conditioning sessions (20 min × two times per day) for two consecutive days. On day 4, mice again underwent the two-compartment test without light stimulation (posttest: 15 min).
**f** Stimulation of MRN serotonergic neurons promoted CPA. The CPA scores calculated as spent time in a light stimulation-associated compartment in the posttest subtracted by that in the pretest in the Venus and CheRiff mice (two-tailed unpaired $t$-test (Venus vs CheRiff), $t_{15} = 3.401$, \*\*$P = 0.0040$, $n = 8$ (Venus) and 9 (CheRiff) mice). **g** Spent time in the compartment associated with light stimulation was significantly decreased after conditioning sessions in the CheRiff mice but not in the Venus control mice (CheRiff: two-tailed paired $t$-test (CheRiff Pre vs CheRiff Post), $t_8 = 4.998$, \*\*$P = 0.0011$, $n = 9$ mice; Venus: two-tailed paired $t$-test (Venus Pre vs Venus Post), $t_7 = 0.4957$, $P = 0.6353$, $n = 8$ mice). Data are presented as mean ± s.e.m. Error bars indicate s.e.m. Source data are provided as a Source Data file.

implanted a small optical fiber targeting the MRN for optogenetic stimulation (Fig. 2d and Supplementary Fig. 2i). Ex vivo electrophysiology experiment revealed that blue light illumination induced firing of CheRiff-expressing cells, which was time-locked to the illumination (80.4 ± 5.5% fidelity (32.2 ± 2.2 action potentials/40 pulses,

$n = 5$ cells from 2 mice, Supplementary Fig. 3h, i). The number of serotonergic neurons expressing c-Fos, a marker of neuronal activity, in CheRiff-expressing animals after 20-min optogenetic stimulation was significantly larger than that in Venus-control animals (Supplementary Fig. 4a–d), confirming CheRiff expression in MRN serotonergic neurons and successful activation of MRN serotonergic neurons with blue light. To assess the contribution of these neurons to aversion-related behavior, we performed a conditioned place aversion (CPA) test (Fig. 2e). The CPA score, used as a proxy of aversion, was defined as the difference in time spent in the optogenetic stimulation-paired chamber (20 Hz) in the posttest session and in the pretest session. The CPA score in the CheRiff-expressing animals was significantly lower than that in the Venus mice (Fig. 2f and Supplementary Fig. 2g). Consistent with this finding, the time spent in the optogenetic stimulation-paired chamber in the posttest session was significantly shorter than that in the pretest session in the CheRiff mice but not in the Venus mice (Fig. 2g), indicating that activation of MRN serotonergic neurons in a neutral context resulted in subsequent avoidance of this context. We then investigated the effect of brief activation of MRN serotonergic neurons on the processing of appetitive stimuli (Figs. 3 and 4). In the two-bottle choice test, the mice were presented with two drinking bottles filled with a 10% sucrose solution. Consumption of the reward from only one of the two bottles triggered optogenetic stimulation (Fig. 3a–d, Supplementary Fig. 5a). Sucrose solution intake from the bottle associated with optogenetic stimulation (20 Hz) was significantly lower than that without stimulation in the CheRiff-expressing mice but not in the Venus-control mice (Fig. 3c), whereas the total intake did not differ between the two groups (Fig. 3d). These results indicated that the activation of MRN serotonergic neurons was aversive.

Although very simple behavior was required in the above tests, performance in these tests was still affected by learning or planning processes. To address this issue, we examined the effect of optogenetic stimulation of MRN serotonergic neurons on anticipatory responses, which develop as a result of learning or planning, and consummatory responses to an automatically delivered reward in a head-fixed setup (Fig. 3e–i and Supplementary Fig. 5b–d). In the pretest session (0–10 min; Pre) and posttest session (20–30 min; Post), mice received 2 μL of 10% sucrose solution every 10 seconds without optogenetic stimulation, and the number of licks per 1 s was recorded. In the stimulation session (10–20 min; Stim), sucrose solution was delivered as in the pretest and posttest sessions, but blue light was applied to the MRN for 2 s from the onset of the sucrose solution delivery. The application of blue light (20 Hz) significantly reduced the frequency of licking behavior during the Stim and Post phases compared to the Pre phase only in the CheRiff mice. The reduction in consummatory licking appeared even in the first trial, suggesting that the activation of MRN serotonergic neurons could suppress the consumption of reward in a learning and planning process-independent manner. No significant differences in the frequency of licking behavior across phases were observed in the Venus mice. Another potential concern was that the nonselective expression of CheRiff might have affected the observed results because approximately 10% of CheRiff-expressing neurons were speculated to be nonserotonergic (Fig. 1d). To address this concern, we performed a similar experiment in Tph2-tTA::tetO-ChR2(C128S)-eYFP bigenic mice[40] where virtually all C128S-expressing cells are serotonergic (Fig. 4a–c, Supplementary Fig. 5e)[41]. We found that blue light application over the MRN significantly reduced the frequency of licking behavior during the Stim phase compared to the Pre phase at −4 s, −2 s, and 0 s. We also found a significantly higher frequency of licking behavior during the Post phase than during the Stim phase at 0 s (Fig. 4a–c). The reduction of licking behavior looks more prominent in the bigenic mice than in CheRiff mice. As we injected AAV into the rostral part of the MRN in CheRiff mice, it is likely that serotonergic neurons in the caudal part of the MRN were more faithfully manipulated in bigenic mice.

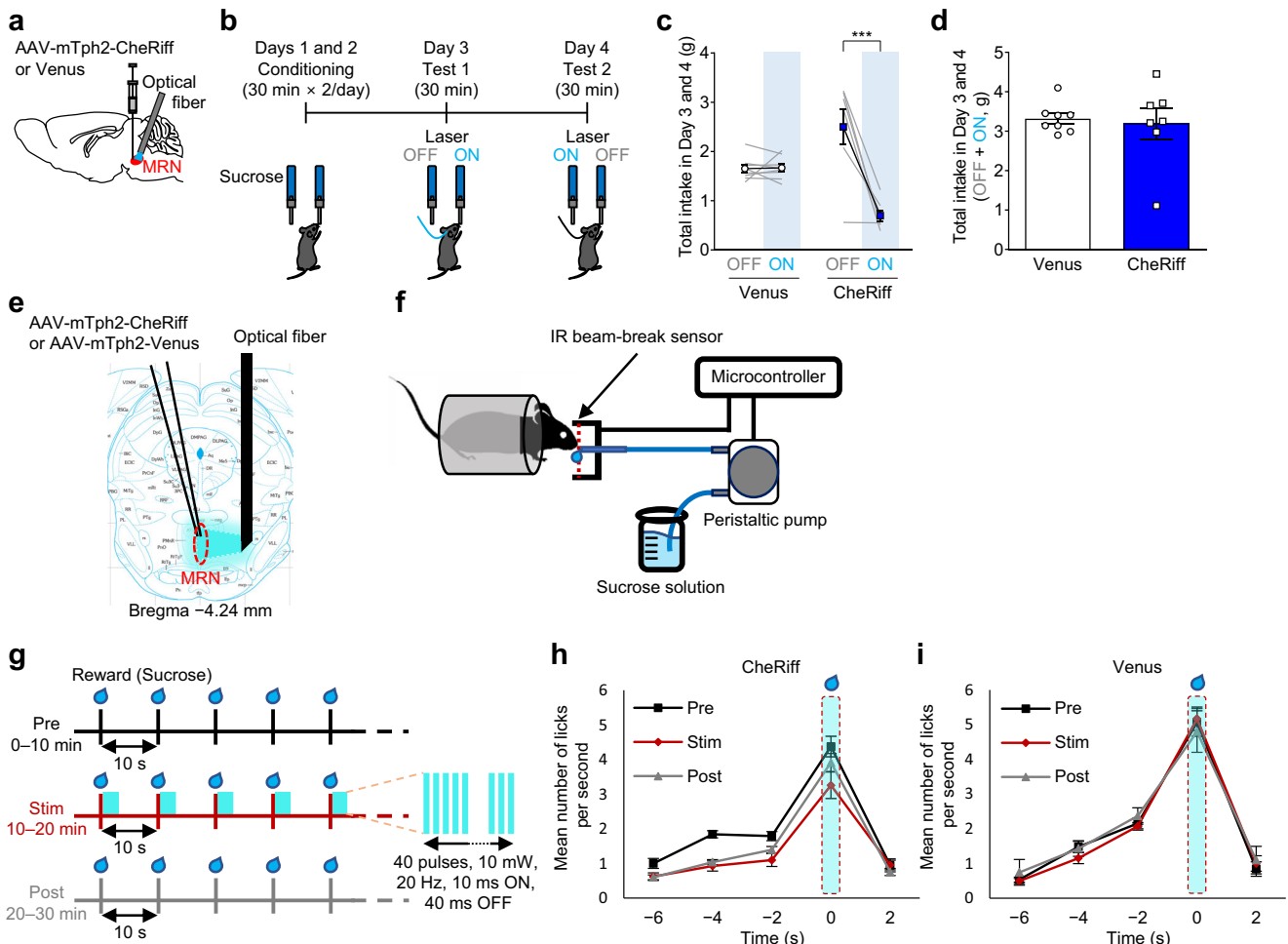

**Fig. 3 | Effect with activation of MRN serotonergic neurons on reward consumption. a** Schematic representation of AAV injection and fiber implantation. **b** Two-bottle choice test. The position of the bottle paired with light was counterbalanced between days 3 and 4. **c** Total intake of sucrose solution in each bottle. Two-way repeated-measures ANOVA with Sidak *post hoc* test, virus × light interaction $F_{1,13} = 26.97$, $P = 0.0002$, virus $F_{1,13} = 0.09712$, $P = 0.7603$, light $F_{1,13} = 26.01$, $P = 0.0002$; *post hoc* test: Valued condition OFF-CheRiff vs ON-CheRiff ***$P < 0.001$, $n = 8$ (Venus) and 7 (CheRiff) mice. **d** Total intake of solution from both of bottles. Two-tailed unpaired *t*-test with Welch's correction, $t_{7.378} = 0.2949$, $P = 0.7762$, $n = 8$ (Venus) and 7 (CheRiff) mice. **e** Schematic representation of AAV injection and fiber implantation and distance from the bregma. **f** Head-fixed setup. Mice were secured by head clamps attached to the chamber frame. A 10% sucrose reward was

delivered through the needle. **g** Schematic diagram of the optogenetic stimulation. A 2 µL drop of 10% sucrose solution reward was delivered every 10 s. **h, i** Number of licks during each phase of the fixed-time schedule task (Black: Pre, Red: Stim, Gray: Post). Each time point of the graph represents the mean number of licks for 2 s. Three-way ANOVA for repeated measures (Phase effect (Venus) ($n = 6$ mice, $F_{2,18} = 0.22$, $P = 0.81$), Phase effect (CheRiff) ($n = 10$ mice, $F_{2,18} = 7.82$, $P = 0.0036$), Virus × Phase interaction ($F_{2,28} = 3.36$, $P = 0.049$), Virus × Time (−6 to 2 s) interaction ($F_{4,56} = 8.29$, $P = 2.52 \times 10^{-5}$)). Multiple comparisons for phase effect in the CheRiff group (Pre vs Post ($P = 0.012$), Pre vs Stim ($P = 0.016$), Stim vs Post ($P = 0.27$) phase effects. Data are presented as mean ± s.e.m. Error bars indicate s.e.m. Source data are provided as a Source Data file. An illustration of a mouse was obtained from Pixabay.

To more directly assess hedonic responses that do not involve goal-directed behavior, we intraorally infused a drop of sucrose solution and observed taste reactivity[42] and facial expression[43] in Tph2-tTA::tetO-ChR2(C128S)-eYFP bigenic mice (Fig. 4d–h). In this task, the mice did not receive any training, and no predictive cue was paired with the unconditioned stimuli. This design enabled us to purely assess the effects of optogenetic manipulation on hedonic responses. We found that blue light application to the MRN significantly reduced the number of tongue reactions and sweet sucrose-driven facial expressions during the Stim phase compared to the Pre phase (Fig. 4f, h). Tongue reactions were recovered during the Post phase, consistent with Figs. 3f–i, 4a–c, suggesting that the sucrose solution was still appetitive after MRN manipulation. Interestingly, however, the facial expression did not return to the baseline expression in the Post phase, suggesting that the previous MRN manipulation partially reduced the value of the sucrose solution. Collectively, these results indicated that increased activity in MRN serotonergic neurons mediated aversion.

## MRN serotonergic neurons densely innervate the interpeduncular nucleus, ventral hippocampus, and medial habenula

To elucidate the serotonergic pathway regulating the processing of reward and aversive stimuli, we focused on the MRN−interpeduncular nucleus (IPN) pathway because the IPN has been shown to play a key role in regulating nicotine preference[44,45], and anterograde tracing of MRN serotonergic neurons has shown that the IPN is densely innervated by these neurons[46]. To confirm the presence of MRN serotonergic projections transduced by AAV, immunohistochemical labeling of eGFP and serotonin transporter (SERT), a marker for serotonergic axons, was performed in the CheRiff-eGFP-expressing mice. We found fiber-like structures positive for eGFP and SERT in the IPN (Fig. 5a, b). Moreover, fluorescent latex microspheres (Retrobeads) injected into the IPN (Fig. 5e, f) were observed in MRN serotonergic neurons 4 weeks after Retrobeads injection. In addition to the IPN, we found fiber-like structures positive for

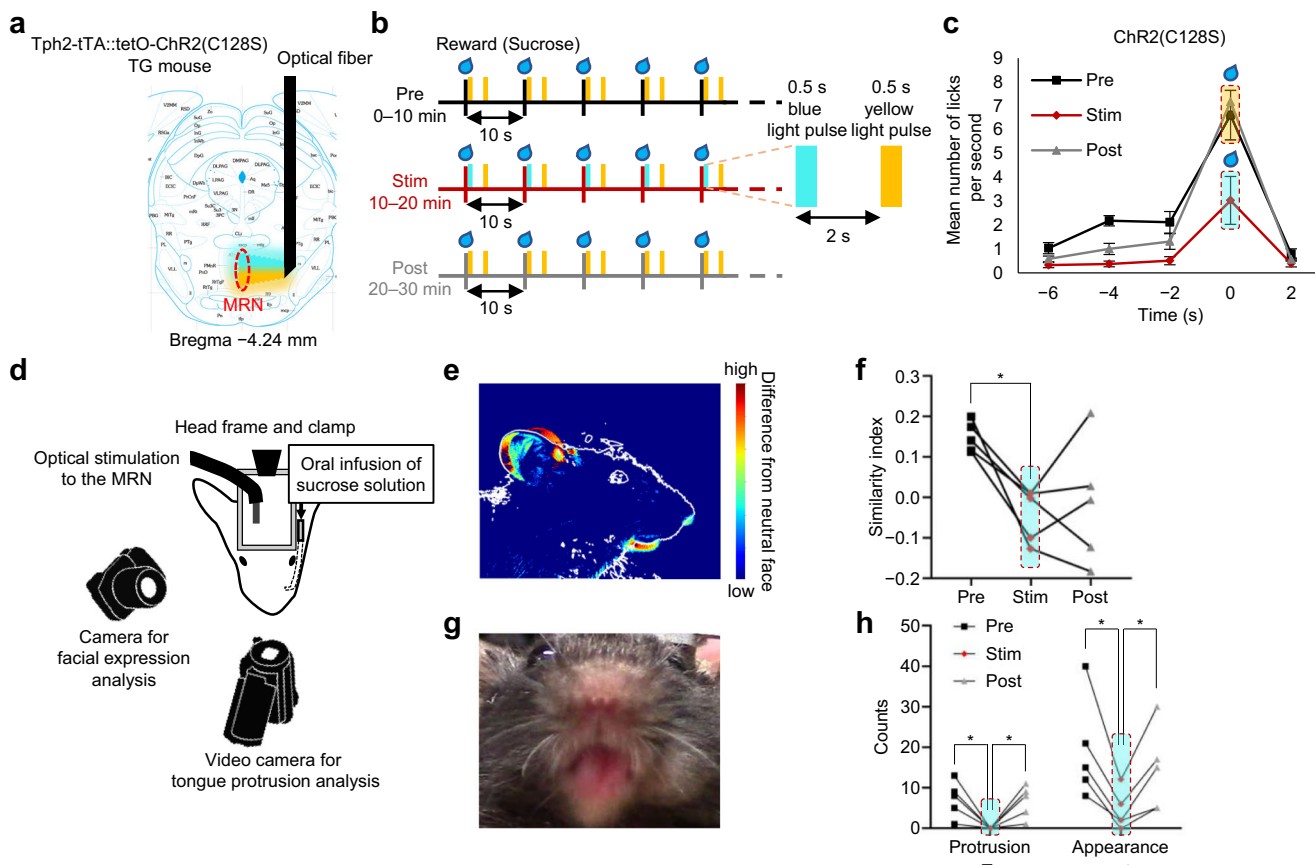

**Fig. 4 | Effect with activation of MRN serotonergic neurons on reward consumption and facial expression in Tph2-tTA::tetO-ChR2(C128S) bigenic mice.**
**a** Schematic representation of fiber implantation and distance from the bregma.
**b** Schematic diagram of the optogenetic stimulation during the fixed-time schedule task. A 2 μL drop of 10% sucrose solution reward was delivered every 10 s. **c** Number of licks during each phase of the fixed-time schedule task (Black: Pre, Red: Stim, Gray: Post). Each time point of the graph represents the mean number of licks for 2 s. A 2-way ANOVA for repeated measures revealed a significant interaction effect of Phase × Time ($F_{8,32} = 5.37$, $P = 0.0003$). Multiple comparisons at each time point revealed a significant Pre vs Stim phase effect in Time −4 s ($P = 0.0059$) and −2 s ($P = 0.015$), and 0 s ($P = 5.76 \times 10^{-7}$). Stim vs Post phase effect was also observed in Time 0 s ($P = 3.25 \times 10^{-8}$). **d** Schematic representation of oral infusion setup in Tph2-

tTA::tetO-ChR2(C128S)-eYFP bigenic mice. **e** Heatmap showing the difference in the face between before and after oral infusion of sucrose solution. **f** Similarity index to prototype face after sucrose infusion. The activation of serotonergic neurons in the MRN reduced the similarity (ANOVA: $F_{1.38,5.50} = 6.53$, $P = 0.041$, two-sided test). **g** A representative photograph showing tongue protrusion after sucrose infusion.
**h**, The activation of serotonergic neurons in the MRN reduced tongue protrusion and appearance (ANOVA for protrusion: $F_{1.70,6.81} = 10.45$, $P = 0.0096$; ANOVA for appearance: $F_{1.28,5.10} = 14.57$, $P = 0.01$). *$P < 0.05$, Tukey's multiple comparisons. A two-sided test was used for statistical analyses. Each point indicates data from one animal. $n = 5$ mice. Data are presented as mean ± s.e.m. Error bars indicate s.e.m. Source data are provided as a Source Data file. An illustration of a camera was obtained from the silhouetteAC.

eGFP in the hippocampus and medial habenula (MHb). Previous reports have shown that MHb plays a critical role in the processing of reward and aversion[47–49]. Previous studies have indicated that the ventral hippocampus (vHP), but not the dorsal hippocampus (dHP), is necessary for regulating approach/avoidance behaviors and processing an aversive predatory odor[50,51]. We further performed immunohistochemical labeling of eGFP and SERT in the vHP and MHb of the CheRiff-eGFP-expressing mice. Similar to the IPN, we found fiber-like structures positive for eGFP and SERT in both the vHP (Fig. 5c) and MHb (Fig. 5d). Moreover, fluorescent Retrobeads injected into the vHP (Fig. 5g, h) and MHb (Fig. 5i, j) were observed in MRN serotonergic neurons. To determine whether the activation of MRN serotonergic neurons affected neural activity in brain areas receiving their projection, we quantified c-Fos expression in these areas following optogenetic stimulation of MRN serotonergic neurons. Ninety minutes after optogenetic stimulation (20 Hz, 10-ms ON and 40-ms OFF (20% duty cycle), 20 min), the expression of c-Fos in the IPN, vHP, and MHb was quantified in the mice expressing CheRiff or Venus in the MRN (Supplementary Fig. 6). We found significantly more c-Fos-positive cells in the CheRiff mice than in the Venus mice in

the IPN but not in the vHP and MHb (IPN: $n = 6$ (Venus) and $n = 8$ (CheRiff), $t_{12} = 2.329$, *$P = 0.0382$, Supplementary Fig. 6a–c; vHP: $n = 6$ (Venus) and $n = 9$ (CheRiff), $t_{13} = 0.7657$, $P = 0.4575$, Supplementary Fig. 6d–f; MHb: $n = 6$ (Venus) and $n = 8$ (CheRiff), $t_{12} = 0.6630$, $P = 0.5199$, Supplementary Fig. 6i–k). However, when we analyzed c-Fos-positive pyramidal and non-pyramidal neurons in the vHP separately, we found significantly more c-Fos-positive non-pyramidal neurons in CheRiff mice than in Venus mice, whereas there was no significant difference between groups in c-Fos-positive pyramidal neurons (vHP (non-pyramidal): $n = 6$ (Venus) and $n = 9$ (CheRiff), $t_{13} = 3.806$, **$P = 0.0022$, Supplementary Fig. 6g; vHP (pyramidal): $n = 6$ (Venus) and $n = 9$ (CheRiff), $t_{13} = 0.4284$, $P = 0.6754$, Supplementary Fig. 6h). These results indicate that optogenetic activation of MRN serotonergic neurons increased neuronal activity in the IPN and vHP.

**Appetitive stimuli inhibit MRN serotonergic neurons projecting to the IPN and vHP, and their inhibition is rewarding**
To measure the changes in the activity of axon terminals of MRN serotonergic neurons, axon-GCaMP6s, a variant of GCaMP6s with enhanced transport to axon terminals[52], was expressed in MRN

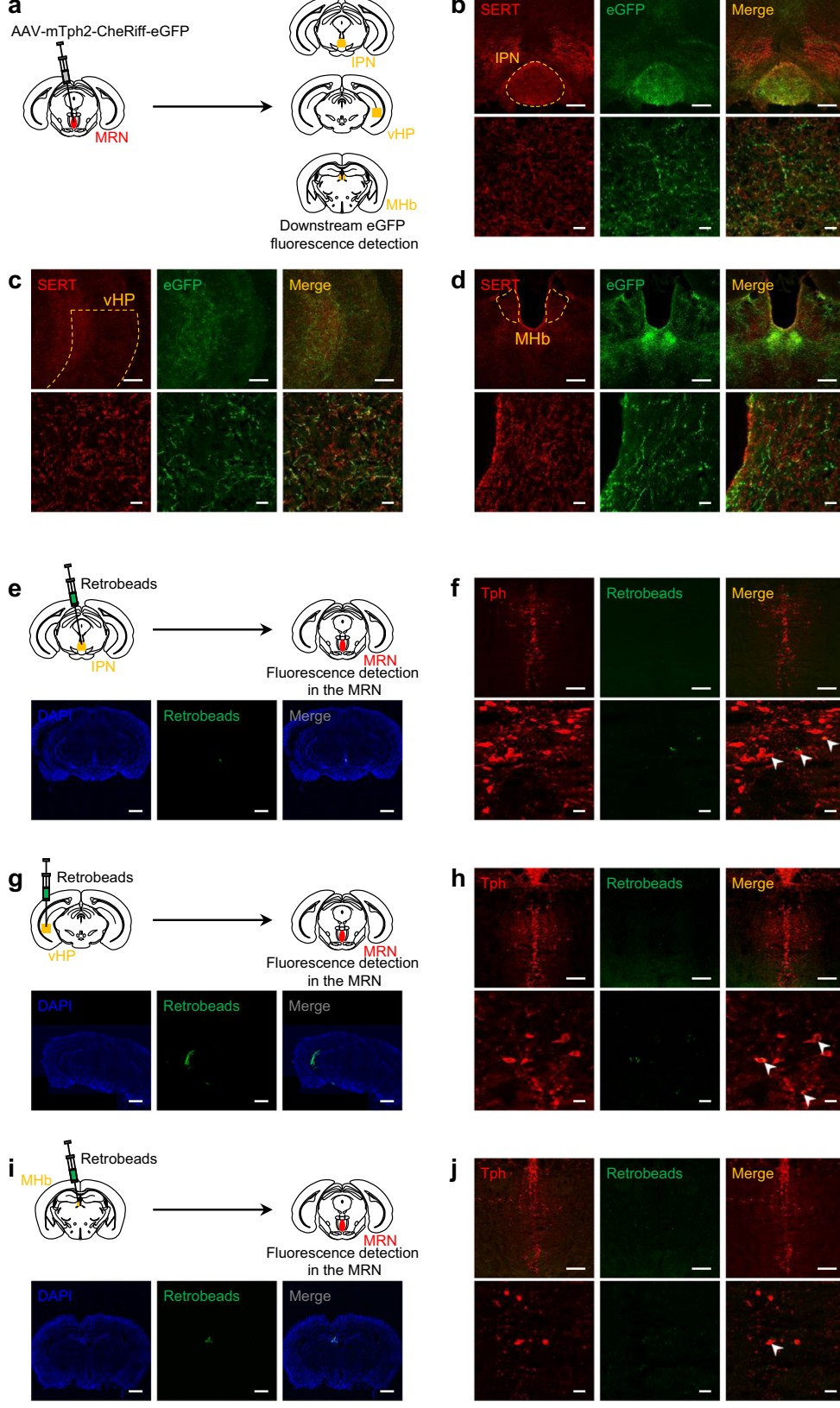

**Fig. 5 | Identification of serotonergic projections from the MRN.**
**a**–**d** Anterograde labeling of serotonergic projections from the MRN in IPN (**b**), vHP (**c**), and MHb (**d**) in AAV-mTph2-CheRiff-eGFP injected mice. SERT serotonin transporter. Scale bars = 200 μm (top) and 20 μm (bottom). **e**–**j** Mice were micro- infused with Retrobeads into the IPN (**e**, **f**), vHP (**g**, **h**), or MHb (**i**, **j**). Retrogradely labeled neurons in the MRN from IPN (**f**), vHP (**h**), or MHb (**j**) were visualized. Arrowheads = colocalized cells. Scale bars = 1 mm (**e**, **g**, **i**), 200 μm (top, **f**, **h**, **j**), and 20 μm (bottom, **f**, **h**, **j**).

serotonergic neurons. After AAV injection into the MRN, an optical fiber was implanted in the IPN (Fig. 6a and Supplementary Figs. 7a and 8a, b), vHP (Fig. 6h and Supplementary Figs. 7g and 8a, c) and MHb (Fig. 6o and Supplementary Figs. 7m and 8a, d) to record axon-GCaMP fluorescence in MRN serotonergic projections to the IPN, vHP, and MHb, respectively. We found that axon-GCaMP fluorescence in the IPN significantly decreased during sucrose consumption (Fig. 6b–d), similar to the results in MRN cell bodies (Fig. 1i). The axon-GCaMP fluorescence in the vHP also decreased during sucrose consumption but to a smaller extent than that in the IPN or MRN (Fig. 6i–k). The extent of fluorescence changes in the IPN and vHP in the axon-GCaMP6s-expressing animals was significantly larger than that in the Venus control animals (IPN: $n = 7$ mice per group, $t_{7.895} = 4.302$, **$P = 0.0027$; vHP: $n = 6$ mice per group, $t_{10} = 4.228$, **$P = 0.0018$; Supplementary Fig. 7b–e, h–k). In contrast, the size of the fluorescence changes in the MHb in the axon-GCaMP6s-expressing mice was similar to that in the Venus mice ($n = 6$ mice per group, $t_{5.561} = 1.458$, $P = 0.1990$; Supplementary Fig. 7n–q), although we found a slight but significant decrease in fluorescence 2–3 s after lick onset from baseline in the axon-GCaMP6s-expressing mice (Fig. 6p–r). Moreover, we measured axon-GCaMP fluorescence in the lateral habenula (LHb) and the paraventricular nucleus of the thalamus (PVT), where clear projections of MRN serotonergic neurons were observed (Fig. 5d). However, we did not find significant fluorescence changes in either nucleus in response to sucrose consumption compared with the Venus control (Supplementary Fig. 7s–v). To assess whether these changes in activity in serotonergic projections from the MRN were causally involved in the regulation of reward-related behavior, we examined the effect of inhibition of these projections in the CPP test. We injected AAV-mTph2-eArchT or AAV-mTph2-Venus into the MRN and implanted an optical fiber into the IPN, vHP or MHb (Fig. 6e, l, s, and Supplementary Fig. 8e–h). Optogenetic inhibition in the IPN and vHP during conditioning sessions significantly increased CPP scores (Fig. 6f, m and Supplementary Fig. 7f, l) and the time spent in the chamber associated with optogenetic inhibition (Fig. 6g, n). In contrast, optogenetic inhibition in the MHb did not significantly affect CPP scores or the time spent in the chamber associated with optogenetic inhibition (Fig. 6t, u and Supplementary Fig. 7r). These data indicated a critical and moderate role of the 5-HT$^{MRN \to IPN}$ and 5-HT$^{MRN \to vHP}$ pathways, respectively, in appropriate processing of reward information.

### Aversive stimuli activate MRN serotonergic neurons projecting to the IPN and vHP, and their activation is aversive

We then measured changes in the activity of axon terminals of serotonergic neurons following the presentation of aversive stimuli (Fig. 7a, f, k, and Supplementary Figs. 9a, e, i and 10a–d). Similar to the responses to a reward (Fig. 6), we found that axon-GCaMP fluorescence in the IPN and vHP, but not in the MHb, significantly increased after a tail pinch (Fig. 7b, g, l and Supplementary Fig. 9b, f, j). However, fluorescence changes in the IPN appeared immediately after the aversive stimuli, while those in the vHP appeared 1 to 2 s later. Thus, we speculated that serotonergic terminals in the IPN are primarily involved in the processing of reward and aversive stimuli compared to those in the vHP. The size of fluorescence changes in the IPN and vHP, but not in the MHb, in the axon-GCaMP6s-expressing animals were significantly larger than those in the Venus control mice (IPN: $n = 6$ (Venus) and $n = 7$ (axon-GCaMP) mice, $t_{11} = 4.124$, **$P = 0.0017$; vHP: $n = 6$ mice per group, $t_{10} = 2.437$, *$P = 0.0350$; MHb: $n = 6$ mice per group, $t_{10} = 1.962$, $P = 0.0782$; Supplementary Fig. 9c, g, k). To assess the causal relationship between the increase in activity and the expression of aversion-related behaviors, we expressed CheRiff or Venus in the MRN and implanted an optical fiber in the IPN, vHP or MHb (Fig. 7c, h, m, and Supplementary Fig. 10e–h). In the CPA test, optogenetic stimulation in the IPN and vHP significantly decreased the CPA score (Fig. 7d, i, and Supplementary Fig. 9d, h) and the time spent

in the chamber associated with optogenetic stimulation (Fig. 7e, j), whereas that in the MHb did not (Fig. 7n, o, and Supplementary Fig. 9l). In Tph2-tTA::tetO-ChR2(C128S)-eYFP bigenic mice where virtually all ChR2-expressing neurons were serotonergic, we found significantly more c-Fos-positive cells in IPN of the bigenic mice than control mice after optogenetic stimulation in the MRN ($n = 5$ (control) and $n = 4$ (bigenic mice), $t_7 = 4.21$, **$P = 0.004$, Supplementary Fig. 9m). Additionally, optogenetic stimulation in the IPN of the bigenic mice significantly decreased the CPA score and the time spent in the chamber associated with optogenetic stimulation ($n = 7$ (control) and $n = 7$ (bigenic mice), (CPA score) $t_{12} = 2.686$, *$P = 0.020$, (spent time-ChR2 bigenic) $t_6 = 4.212$, **$P = 0.0056$, Supplementary Fig. 9n, o). Considering the spatial closeness between IPN and MRN and the presence of serotonergic fibers passing through the IPN, optogenetic stimulation in the IPN may affect serotonergic projections from the MRN to other brain regions, including vHP, which leads to an apparently strong effect induced by IPN stimulation. Therefore, we performed optogenetic stimulation of 5-HT$^{MRN \to IPN}$ and counted c-Fos-expressing cells in the IPN and vHP to determine the extent to which 5-HT$^{MRN \to IPN}$ stimulation affects serotonergic terminals in the vHP. Stimulation significantly increased the number of c-Fos-expressing cells in the IPN, but did not affect the number of c-Fos-positive non-pyramidal neurons in the vHP (Supplementary Fig. 6l, m, o, p, q). Furthermore, there was no significant increase in c-Fos-expressing MRN serotonergic neurons after optogenetic stimulation with 5-HT$^{MRN \to IPN}$ (Supplementary Fig. 6n). These results indicate successful stimulation of 5-HT$^{MRN \to IPN}$ with minimal intervention to other serotonergic projections, including 5-HT$^{MRN \to vHP}$. Collectively, these data suggested that 5-HT$^{MRN \to IPN}$, and 5-HT$^{MRN \to vHP}$ neurons play a central role in the processing of aversive information.

### Activation of 5-HT receptors, including 5-HT$_{2A}$ receptors in the IPN, is necessary for the aversive effect of MRN serotonergic neuron activation

To investigate which 5-HT receptor subtypes mediate serotonergic signaling from the MRN, we examined the effect of a 5-HT receptor antagonist. Previous reports have shown that antagonists of 5-HT$_{1A}$ and 5-HT$_{2A}$ receptors mitigate freezing behavior[53,54] induced by electrical shock and drug-induced place aversion[55]. Accordingly, we intraperitoneally administered WAY-100635 (0.5 mg/kg; 5-HT$_{1A}$ receptor antagonist), MDL-100907 (0.5 mg/kg; 5-HT$_{2A}$ receptor antagonist), or vehicle (saline) 30 min before the conditioning session in the CPA test (Fig. 8a, b and Supplementary Fig. 11b). In vehicle-treated mice, optogenetic stimulation in the MRN significantly decreased the CPA score, whereas pretreatment with WAY-100635 or MDL-100907 almost completely abolished this effect (Fig. 8c and Supplementary Fig. 11a). Similarly, optogenetic stimulation significantly decreased the time spent in the chamber associated with stimulation in the mice pretreated with the vehicle but not in the mice pretreated with WAY-100635 or MDL-100907 (Fig. 8d). To identify the brain region responsible for these effects, we then locally administered the antagonists. We found that the neural activity of the 5-HT$^{MRN \to IPN}$ pathway promptly and more prominently changed in responses to rewarding and aversive stimuli compared to that of the 5-HT$^{MRN \to vHP}$ pathway. Therefore, we evaluated the effects of intra-IPN administration of the antagonists (Fig. 8e). Five minutes before the conditioning session in the CPA test, we administered WAY-100635 (1 µg in 0.2 µL), MDL-100907 (0.1 µg in 0.2 µL) or a vehicle (saline, 0.2 µL) through injection cannula implanted into the IPN (Fig. 8f and Supplementary Fig. 11d). We found that intra-IPN administration of MDL-100907, but not of WAY-100635, reversed the reduction in the CPA score induced by the activation of the MRN (Fig. 8g and Supplementary Fig. 11c). Moreover, optogenetic stimulation significantly decreased the time spent in the chamber associated with stimulation in vehicle- or WAY-100635-treated mice but not in MDL-100907-

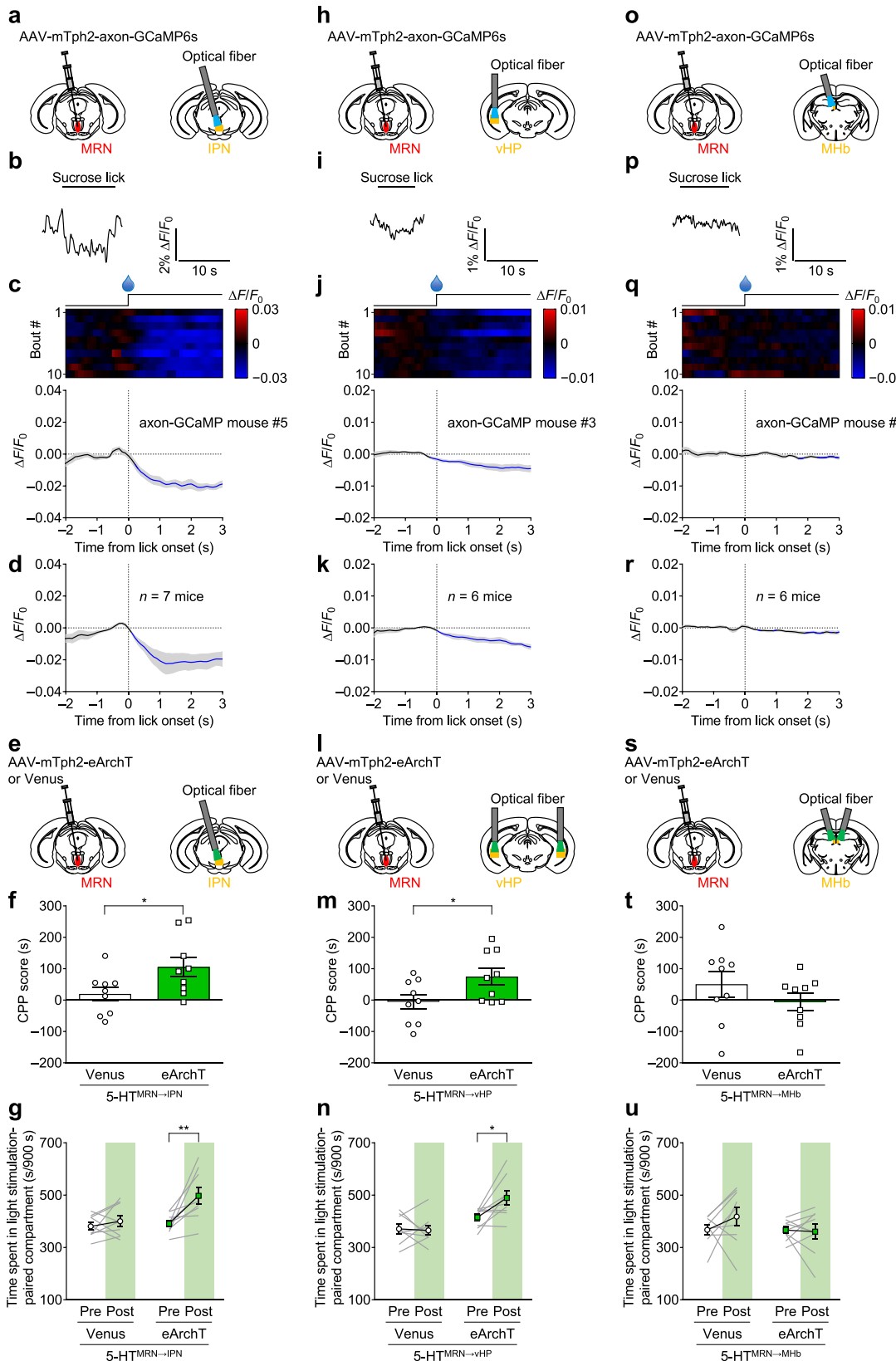

treated mice (Fig. 8h). Immunohistochemical analysis using two different 5-HT$_{2A}$ receptor antibodies revealed immunoreactivity with cell soma-like morphology in the IPN (Supplementary Fig. 11g, h). We also found 5-HT$_{2A}$ mRNA expression in the IPN using RT-PCR (Supplementary Fig. 11i). We examined the effects of intra-IPN administration of TCB-2, a 5-HT$_{2A}$ agonist, on CPA scores without

optogenetic manipulation. We found that intra-IPN administration of TCB-2 (0.05 µg/0.5 µL) significantly decreased the CPA score and the time spent in the chamber associated with TCB-2 (Fig. 8i, j and Supplementary Fig. 11e, f). Taken together, these results indicate that 5-HT receptors, including 5-HT$_{2A}$ receptors in the IPN, are involved in the processing of aversive stimuli.

**Fig. 6 | Projection-specific responses of the MRN serotonergic neurons to rewarding stimuli and effects with their inhibition in the CPP test.**
**a, e, h, l, o, s** Schematic representation of AAV injection and fiber implantation. **b, i, p** Representative raw trace of axon-GCaMP fluorescence in the IPN (**b**), vHP (**i**), and MHb (**p**). **c, j, q** Top: heatmap of signals (red–blue, high–low). One licking bout per row. Bottom: averaged axon-GCaMP6s signals of one mouse in the IPN (**c**), vHP (**j**), and MHb (**q**). $n = 10$ bouts. **d, k, r** Mean axon-GCaMP6s signals for 7 (IPN) or 6 (vHP, MHb) mice in the IPN (**d**), vHP (**k**), and MHb (**r**). Lines and shaded areas indicate mean and s.e.m., respectively. Blue segments indicate a statistically significant increase ($P < 0.05$; permutation test). **f, m, t** Inhibition of 5-HT$^{MRN \to IPN}$ and 5-HT$^{MRN \to vHP}$ promoted CPP. The CPP scores calculated as spent time in a light stimulation-associated compartment in the posttest subtracted by that in the pretest in the Venus and eArchT mice with fiber implanted in the IPN (**f**), vHP (**m**),

and MHb (**t**) (IPN: $t_{16} = 2.237$, *$P = 0.0399$, $n = 9$ mice; vHP: $t_{16} = 2.260$, *$P = 0.0382$, $n = 9$ mice; MHb: $t_{16} = 1.141$, $P = 0.2705$, $n = 9$ mice). **g, n, u**, Spent time in the compartment associated with light stimulation in the IPN (**g**) and vHP (**n**) but not MHb (**u**) was significantly increased after conditioning sessions in the eArchT mice (IPN-eArchT: (Pre vs Post), $t_8 = 3.486$, **$P = 0.0082$, $n = 9$ mice; IPN-Venus: (Pre vs Post), $t_8 = 0.8922$, $P = 0.3983$, $n = 9$ mice; vHP-eArchT: (Pre vs Post), $t_8 = 2.800$, *$P = 0.0232$, $n = 9$ mice; vHP-Venus: (Pre vs Post), $t_8 = 0.2371$, $P = 0.8186$, $n = 9$ mice; MHb-eArchT: (Pre vs Post), $t_8 = 0.2155$, $P = 0.8347$, $n = 9$ mice; MHb-Venus: (Pre vs Post), $t_8 = 1.233$, $P = 0.2527$, $n = 9$ mice). Two-tailed unpaired $t$-test (**f, m, t**) or two-tailed paired $t$-test (**g, n, u**) were used for statistical analysis. Data are presented as mean ± s.e.m. Error bars indicate s.e.m. Source data are provided as a Source Data file.

## Discussion

Serotonin, a monoaminergic neurotransmitter, has long been associated with a variety of brain functions, including the processing of reward/aversive information, mood regulation, and memory[56,57]. Our study identified and clarified the critical role of MRN serotonergic neurons and their projections to the IPN in the processing of reward and aversive information through cell type-specific activity recordings and optogenetic manipulation. Several previous studies have separately indicated that either the MRN or IPN was involved in the processing of stimuli with a negative valence[29–32,44,45]. We replicated these findings using more selective manipulations and more direct measurements, such as taste reactivity and facial expression, which do not involve goal-directed behavior, and further identified the role of serotonergic projections between these two brain regions, the MRN and IPN, in the processing of reward and aversive information. Although we selectively recorded and manipulated MRN serotonin neurons only, our results were similar to those of vGluT2-expressing MRN glutamatergic neurons[32]. Further studies are required to determine whether the roles of serotonin neurons in the MRN could be discriminated from the roles of vGluT2-expressing MRN glutamatergic neurons. Our results identified the 5-HT$^{MRN \to IPN}$ pathway as a crucial circuit for the appropriate processing of reward and aversive information. We revealed that aversive stimuli increased MRN serotonergic neuron activity and that optogenetic activation of MRN serotonergic neurons increased the number of c-Fos-positive cells in the IPN and non-pyramidal neurons in the vHP. Additionally, we found that stimulation of 5-HT$^{MRN \to IPN}$ significantly increased the number of c-Fos-expressing cells in the IPN, but did not affect the number of c-Fos-positive non-pyramidal neurons in the vHP and c-Fos-positive cells in the MRN. These results indicate successful stimulation of 5-HT$^{MRN \to IPN}$ with minimal intervention in 5-HT$^{MRN \to vHP}$. Nevertheless, we cannot rule out the possibility that stimulation of 5-HT$^{MRN \to IPN}$ may affect other serotonergic projections from the MRN, indicating the importance of brain-wide activity mapping for unbiased identification of downstream targets of MRN serotonergic neurons in the future. Moreover, we cannot rule out the possibility that 5-HT$^{MRN \to PVT}$ and 5-HT$^{MRN \to LHb}$ play a critical role in the processing of aversive information in addition to 5-HT$^{MRN \to IPN}$, while our data indicate these projections did not respond to reward. Furthermore, previous reports have revealed that LHb and PVT play an important role in the processing of aversive stimuli[58,59]. Therefore, further experiments to investigate the role of 5-HT$^{MRN \to PVT}$ and 5-HT$^{MRN \to LHb}$ in the processing of aversive information, such as morphine withdrawal, will be of high importance. Previous studies have shown that acute nicotine withdrawal increases neural activity in the IPN, and activation of IPN neurons projecting to the laterodorsal tegmentum (LDTg) results in aversion-related place avoidance[45,60]. Therefore, it is possible that therapeutics modulating activity of the 5-HT$^{MRN \to IPN}$ pathway may be useful in the treatment of withdrawal syndromes resulting from the cessation of addictive drug administration, although the projection patterns of IPN neurons activated by MRN serotonergic neurons have yet to be determined. Moreover, we

showed the involvement of 5-HT receptors, including 5-HT$_{2A}$ receptors in the IPN, in the acquisition of CPA. Interestingly, previous studies have shown that inverse agonizts of the 5-HT$_{2A}$ receptor mitigate withdrawal symptoms in chronically nicotine-treated rats[61,62]. Taken together with the abovementioned report showing that neural activity in the IPN is increased by acute nicotine withdrawal, it is possible that IPN neurons expressing 5-HT$_{2A}$ receptors play a critical role in the expression of withdrawal symptoms of nicotine and its treatment. In addition to 5-HT$_{2A}$ receptors, previous reports have shown strong expression of several 5-HT receptors, including 5-HT$_{1A}$[63], 5-HT$_{1B}$[64], and 5-HT$_4$[65]. Considering the limitation of pharmacological intervention with regard to receptor selectivity, further analysis using gene knockdown and Cas9-mediated in vivo gene knockout[66] is necessary for determining the 5-HT receptors critical for processing aversive stimuli.

Appetitive stimuli suppressed the activity of MRN serotonergic neurons (Fig. 1f–i), while they activated DRN serotonergic neurons[16,17] (Supplementary Fig. 12a–h). We and others have shown that activation of DRN serotonergic neurons produces reward-related behaviors mainly through its terminals in the ventral tegmental area (VTA)[14,15]. These reports indicated that the activity of DRN serotonergic neurons plays a role in reward signaling. Anatomical studies have demonstrated that afferents from the IPN are found in the LHb[67]. Lammel et al. have shown that LHb inputs to the VTA preferentially activate mesoprefrontal dopaminergic neurons, whose activation is aversive[68]. Therefore, it is possible that activation of the 5-HT$^{MRN \to IPN}$ pathway by aversive stimuli activates the LHb-VTA-medial prefrontal cortex pathway, which codes for a negative value. Moreover, a previous study showed that activation of LDTg-projecting IPN neurons induced inhibition of VTA-projecting LDTg neurons and place avoidance[45]. Interestingly, mesolimbic dopaminergic neurons are innervated by LDTg neurons, whereas activation of VTA-projecting LDTg neurons induced place preference[68]. Therefore, it is also possible that activation of the 5-HT$^{MRN \to IPN}$ pathway by aversive stimuli inhibits the LDTg-VTA-nucleus accumbens (NAc) pathway, which codes for a positive value. On the other hand, vesicular glutamate transporter 3 (vGluT3)-expressing DRN neurons, which also contain serotonin, strongly innervate mesolimbic dopaminergic neurons[69]. Wang et al. revealed that activation of VTA-projecting DRN serotonergic neurons increases extracellular dopamine levels in the NAc and induces a place preference[70]. Taken together, activation of DRN serotonergic neurons by rewarding stimuli activates the VTA-NAc pathway, which codes positive value. Collectively, it is possible that signaling from the MRN and DRN converges in mesolimbic dopaminergic neurons in the VTA, while signaling from the MRN also regulates mesoprefrontal dopaminergic neurons in the VTA, although further investigation is necessary to determine to what extent the activation/inhibition of MRN serotonergic neurons affects the activity of mesolimbic and mesoprefrontal dopaminergic neurons. SSRIs, which upregulate serotonergic neurotransmission from both the MRN and DRN and thus enhance these discrete serotonergic systems coding for opposite values, may lead to less efficacy on anhedonia

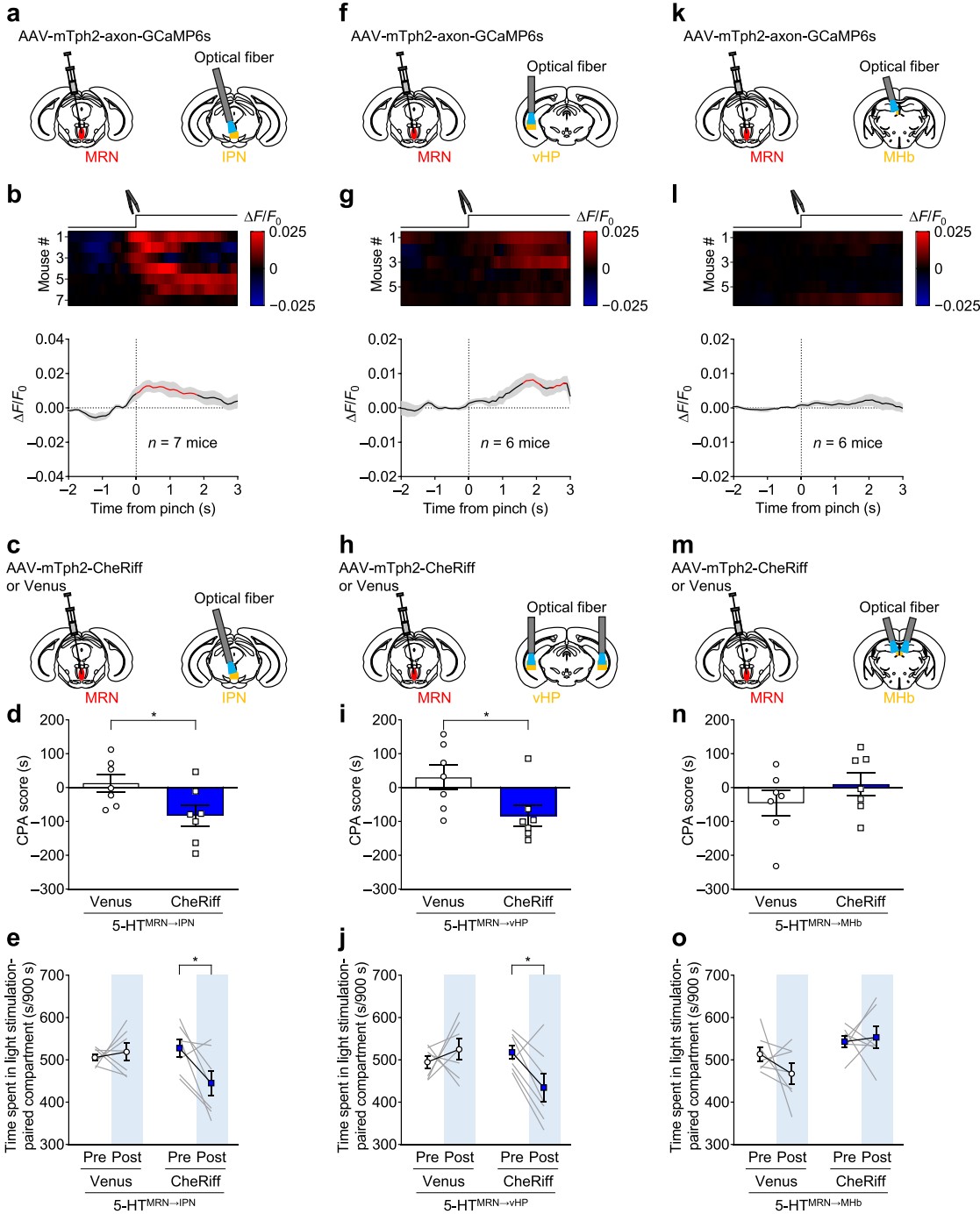

**Fig. 7 | Aversive stimuli activate serotonergic neurons from the MRN to IPN and vHP, and their activation elicits CPA. a**, **f**, **k** Schematic representation of AAV injection and fiber implantation. **b**, **g**, **l** Top: heatmap of axon-GCaMP6s signals in the IPN (**b**), vHP (**g**), and MHb (**l**) (red–blue, high–low) for each trial (one trial per mouse). Each row represents the data of one mouse. $n = 7$ (IPN) or 6 (vHP and MHb) mice. Bottom: averaged traces of axon-GCaMP6s signals in the IPN (**b**), vHP (**g**), and MHb (**l**). Lines and shaded areas indicate mean and s.e.m., respectively. The red segment indicates a statistically significant increase from the baseline ($P < 0.05$; permutation test). **c**, **h**, **m** Schematic representation of AAV injection and fiber implantation. **d**, **i**, **n** Activation of 5-HT$^{MRN \to IPN}$ and 5-HT$^{MRN \to vHP}$ pathways promoted CPA. The CPA scores calculated as spent time in light stimulation-associated compartment in the posttest subtracted by that in the pretest in the Venus and CheRiff mice with fiber implanted in the IPN (**d**), vHP (**i**), and MHb (**n**) (IPN: $t_{12} = 2.367$, *$P = 0.0356$, $n = 7$ mice; vHP: $t_{12} = 2.407$, *$P = 0.0331$, $n = 7$ mice; MHb: $t_{12} = 1.146$, $P = 0.2742$, $n = 7$ mice). **e**, **j**, **o** Spent time in the compartment associated with light stimulation in the IPN (**e**) and vHP (**j**) but not MHb (**o**) was significantly decreased after conditioning sessions in the CheRiff mice (IPN-CheRiff: (Pre vs Post), $t_6 = 2.642$, *$P = 0.0385$, $n = 7$ mice; IPN-Venus: (Pre vs Post), $t_6 = 0.5121$, $P = 0.6268$, $n = 7$ mice; vHP-CheRiff: (Pre vs Post), $t_6 = 2.756$, *$P = 0.0330$, $n = 7$ mice; vHP-Venus: (Pre vs Post), $t_6 = 0.8309$, $P = 0.4378$, $n = 7$ mice; MHb-CheRiff: (Pre vs Post), $t_6 = 0.3089$, $P = 0.7678$, $n = 7$ mice; MHb-Venus: (Pre vs Post), $t_6 = 1.259$, $P = 0.2549$, $n = 7$ mice. Two-tailed unpaired $t$-test (**d**, **i**, **n**) or two-tailed paired $t$-test (**e**, **j**, **o**) were used for statistical analysis. Data are presented as mean ± s.e.m. Error bars indicate s.e.m. Source data are provided as a Source Data file.

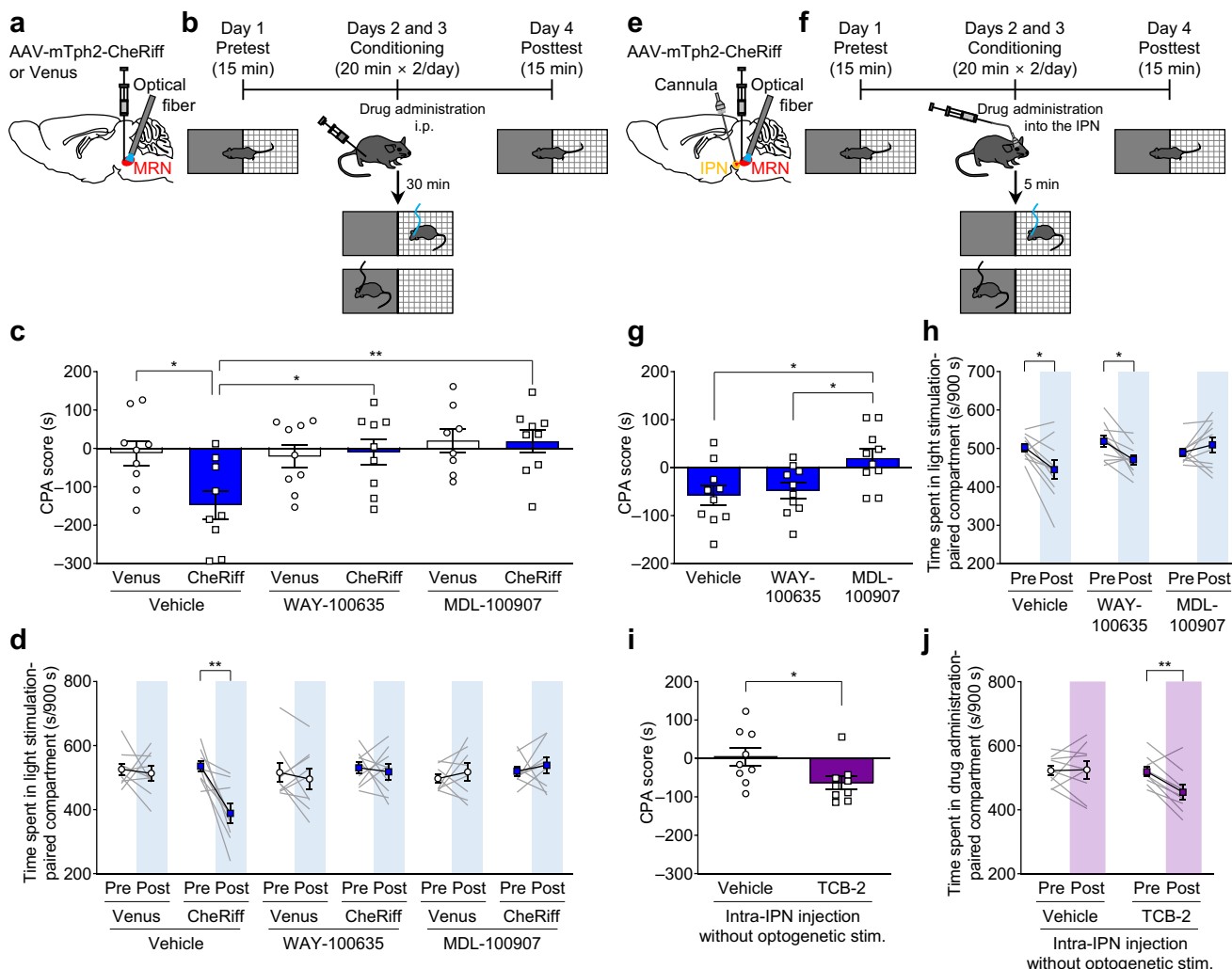

**Fig. 8 | Effect with serotonin receptor antagonists on CPA induced by stimulation of MRN serotonergic neurons and effect with intra-IPN injection of 5-HT$_{2A}$ receptor agonist on CPA. a, e** Schematic representation of experiments. **b** CPA test with pretreatment with serotonin receptor antagonists (WAY-100635 (0.5 mg/kg) or MDL-100907 (0.5 mg/kg)) or vehicle (1% DMSO in saline). **c** CPA scores. Two-way ANOVA with Tukey *post hoc* test, virus × drug interaction $F_{2,47} = 3.215$, *$P = 0.0491$, virus $F_{1,47} = 2.539$, $P = 0.1178$, drug $F_{2,47} = 4.907$, *$P = 0.0116$; *post hoc* test: Venus-Vehicle vs CheRiff-Vehicle *$P = 0.0465$, CheRiff-Vehicle vs CheRiff-WAY-100635 *$P = 0.0401$, CheRiff-Vehicle vs CheRiff-MDL-100907 **$P = 0.0073$. $n = 9$ (except for Venus-MDL-100907) and 8 (Venus-MDL-100907)). **d** Spent time in the compartment associated with light. CheRiff-Vehicle: $t_8 = 3.959$, **$P = 0.0042$, $n = 9$ mice; CheRiff-WAY-100635: $t_8 = 0.3798$, $P = 0.7140$, $n = 9$ mice; CheRiff-MDL-100907: $t_8 = 0.6239$, $P = 0.5501$, $n = 9$ mice. **f** CPA test with intra-IPN injection with serotonin receptor antagonists (WAY-100635 (1 μg in 0.2 μL) or MDL-

100907 (0.1 μg in 0.2 μL)) or vehicle (1% DMSO in saline, 0.2 μL). **g** CPA scores. one-way ANOVA with Tukey *post hoc* test, $F_{2,27} = 5.327$, *$P = 0.0112$; *post hoc* test: Vehicle vs MDL-100907 *$P = 0.0153$, WAY-100635 vs MDL-100907 *$P = 0.0374$, Vehicle vs WAY-100635 $P = 0.9208$. $n = 10$ mice. **h** Spent time in the compartment associated with light. Vehicle: $t_9 = 2.897$, *$P = 0.0177$, $n = 10$ mice; WAY-100635: $t_9 = 3.009$, *$P = 0.0147$, $n = 10$ mice; MDL-100907: $t_9 = 1.053$, $P = 0.3196$, $n = 10$ mice. **i** CPA induced by intra-IPN injection of TCB-2. CPA test was performed as described in (**f**) but without optogenetic stimulation. Two-tailed unpaired *t*-test, $t_{16} = 2.321$, *$P = 0.0338$, $n = 9$ mice. **j** Spent time in the drug-associated compartment. Vehicle: $t_8 = 0.1262$, $P = 0.9027$, $n = 9$ mice; TCB-2: $t_8 = 3.725$, **$P = 0.0058$, $n = 9$ mice. Two-tailed paired *t*-test was used for statistical analyses unless otherwise stated. Data are presented as mean ± s.e.m. Error bars indicate s.e.m. Source data are provided as a Source Data file.

in major depression. From this perspective, differentially expressed genes between MRN and DRN serotonergic neurons[71,72] will be an ideal drug target to treat major depression through pathway-specific modulation of serotonergic neurotransmission. Through analysis of MRN GCaMP fluorescence in the same baseline setting as that of a previous report on DRN serotonergic neurons[16], we found increased GCaMP fluorescence before sucrose licking. As the tested mice had experienced a number of sucrose licks before measurement to acclimate to the setup, further measurement using appetitive Pavlovian conditioning, as previously reported in the DRN[17], is needed to determine whether increased activity in MRN serotonin neurons is associated with reward anticipation.

Previous reports have shown that MRN neurons play a critical role in the regulation of hippocampal theta oscillations, locomotion, and

memory function[57,73–76], as well as in the processing of appetitive and aversive stimuli. Although the extent of suppression was much smaller than that of sucrose, we found that immobility induced a significant decrease in GCaMP fluorescence in the MRN. Similarly, the initiation of locomotion induced a significant increase in GCaMP fluorescence in the MRN, although the size of the increase was substantially smaller than that of the tail pinch (Supplementary Fig. 1m, n and 3a, b). Additionally, quinine, another aversive stimulus[16,17], blunted water-induced decreases in GCaMP fluorescence in MRN. Furthermore, auditory cues associated with foot shocks, which induce immobility in a new context[38], increased GCaMP fluorescence in the MRN. Collectively, these results indicate that MRN serotonergic neurons respond to rewarding and aversive stimuli irrespective of locomotor changes. At the same time, our data indicate that locomotor activity affects the

activity of MRN serotonergic neurons, although the size of the changes was relatively small. This could stem from the number of neurons responding to each stimulus, as fiber photometry measures the fluorescence of a whole nucleus as a proxy for population activity. Therefore, further analysis using in vivo endoscopy is required to determine whether locomotor changes robustly affect the activity of a subpopulation of MRN serotonergic neurons. Additionally, we have to note that it is possible that the difference in the size of $Ca^{2+}$ response to the initiation of locomotion and tail-pinch may be due to the difference in locomotor speed because we did not align the $Ca^{2+}$ signals to a change of locomotor speed. Although a recent report has indicated that hippocampal theta oscillations are mainly modulated by non-serotonergic MRN neurons[77], it is possible that the manipulation of MRN serotonergic neurons affects memory formation. However, we found that activation and inhibition induced strong CPA and CPP, both of which required intact memory function. These results indicate that mice have sufficient memory to associate the conditioned chamber with valence, even in the presence of manipulation of MRN serotonergic neurons. We also found that activation of MRN serotonin neurons reduced consummatory licking in the head-fixed setup and hedonic response in facial expression analysis, both of which are much less amenable to confounding memory function. Although these results strongly support our notion that inhibition and activation of MRN serotonergic neurons are rewarding and aversive, respectively, we cannot rule out the possibility that the memory-enhancing effect of manipulation of MRN serotonergic neurons overrode CPA and CPP by manipulating MRN serotonergic neurons. Further analysis of the role of MRN serotonin neurons in regulating memory formation and hippocampal theta oscillation will be crucial for a better and more comprehensive understanding of the role of MRN serotonergic neurons.

We found that the activity of serotonergic terminals in the vHP was also altered in response to aversive or reward stimuli, but the alterations were delayed compared to those in the IPN (Figs. 6 and 7). Given that previous studies have shown that the 5-HT$^{MRN \to vHP}$ pathway regulates aversive memory and anxious states[78,79], this pathway might mediate indirect or secondary effects of the processing of aversive stimuli rather than the processing itself. It should also be noted that the 5-HT$^{MRN \to vHP}$ pathway is activated during sustained goal-directed actions, although it is not directly related to the processing of reward and aversive stimuli[80]. Mice might perceive the sustained action as aversive or stressful, although this sustained activity would result in preferable outcomes later. Serotonin 5-HT$_3$ receptors in the ventral hippocampus are necessary for sustaining goal-directed actions[80], while aversive memory and anxiety are regulated by 5-HT$_7$ and 5-HT$_{2C}$ receptors in the vHP, respectively[79,81].

Although we focused on the IPN, vHP, and MHb as MRN-projecting regions, we do not necessarily deny the roles of other MRN-projecting regions in the processing of reward and aversive stimuli. MRN serotonergic neurons project their fibers not only to the vHP but also to the dHP. Luchetti et al. identified two types of serotonergic fibers from the MRN to dHP through two-photon calcium imaging of serotonergic fibers in dHP[80]. They found that the two types respond to either reward or movement and showed that inhibition of the 5-HT$^{MRN \to dHP}$ pathway decreased the time spent in the area associated with reward presentation. In contrast with this report, we demonstrated that 5-HT$^{MRN \to vHP,IPN}$ pathways negatively and positively responded to reward and aversive stimuli, respectively. We speculate that such diversity of roles played by MRN serotonergic neurons could be attributed to different subpopulations in MRN serotonergic neurons, each innervating different brain areas and responding to a reward in opposite ways. Of note, recent single-cell RNA-seq analyses have clearly demonstrated that serotonergic neurons in the DRN and MRN can be subdivided into more than ten clusters[71,72]. Moreover, our AAV infected most of the serotonin neurons present in the MRN (Fig. 1b–d in the present study), while AAV in Luchetti et al. infected a small proportion of serotonin neurons in the MRN (Fig. 2B in Luchetti et al.)[82]. Therefore, it is possible that each of these subpopulations has a distinct projection pattern and differentially responds to a reward.

The MHb plays a central role in regulating reward processing, mood, and decision-making[47,83,84]. Although this nucleus is densely innervated by MRN serotonergic neurons (Fig. 5)[46], no significant effect of the 5-HT$^{MRN \to MHb}$ manipulation was observed in either the CPP or CPA test (Figs. 6 and 7). Previous reports have demonstrated that the activity of MHb neurons is increased by nicotine[85], which leads to an increase in extracellular dopamine levels in the NAc via the activation of nicotinic receptors in the MHb[86], indicating a critical role of MHb in the effect of addictive drugs. Consistent with this notion, intra-MHb administration of an antagonist for nicotinic receptors precipitated withdrawal symptoms in chronically nicotine-treated mice[87]. Interestingly, intra-MHb administration of nicotinic receptor antagonists attenuated morphine self-administration but did not affect responding for sucrose[48]. Therefore, it is possible that manipulating the 5-HT$^{MRN \to MHb}$ pathway may affect reward-related behaviors after chronic treatment with addictive drugs.

Although our results strongly suggest a key role of 5-HT$^{MRN \to IPN}$, there is a limitation of our study, mainly due to the presence of serotonergic fibers passing through the IPN and the spatial closeness between the IPN and MRN. Considering the spatial closeness of the IPN and MRN, it is possible that GCaMP fluorescence in the MRN may affect photometry measurements in the IPN. However, the histochemical analysis revealed that optogenetic manipulation in the IPN with much more intense excitation light than photometry measurements did not affect the activity in the MRN (Supplementary Fig. 6n). Hence, it is less likely that axon-GCaMP fluorescence in the MRN affected axon-GCaMP fluorescence in the IPN. In contrast, we cannot rule out the possibility that the axon-GCaMP fluorescence of serotonergic fibers passing through the IPN affects that in the IPN. However, a previous report has shown that fluorescent changes in axon-GCaMP6 in presynaptic boutons are twice as large as those in the axon shaft, whereas there was no difference in the baseline fluorescence of axon-GCaMP6 in presynaptic boutons and axon shaft[52]. Hence, the larger the contribution of baseline fluorescence derived from the fiber passing through the IPN, the less the contribution of axon-GCaMP fluorescence of presynaptic boutons in the IPN (i.e., 5-HT$^{MRN \to IPN}$) to fluorescence changes. If fluorescence changes in the IPN may largely originate from the axon shaft passing through the IPN, there should be other brain regions where much larger fluorescent changes are observed than in the IPN. However, we found that fluorescence changes (relative to baseline fluorescence ($F_0$)) in the IPN were much larger than those in other brain regions, including vHP, MHb, LHb, and PVT (Figs. 6 and 7, Supplementary Fig. 7s–v), where dense MRN serotonergic projections were observed. Collectively, these results indicate a relatively large contribution of 5-HT$^{MRN \to IPN}$ to the fluorescence changes in the IPN. Nevertheless, we cannot completely rule out the possibility that other MRN serotonergic projections respond to appetitive and aversive stimuli. For retrograde labeling, previous reports have indicated that Retrobeads enable retrograde labeling with minimal entry into undamaged fibers of passage and minimal diffusion at the injection site[88], indicating that retrograde labeling from the IPN to the MRN serotonergic neurons, which we observed here, was mainly due to 5-HT$^{MRN \to IPN}$.

In conclusion, we identified MRN serotonergic neurons, including those projecting to the IPN, as important mediators in the processing of rewards and aversive stimuli. Moreover, our data suggested that 5-HT receptors, including 5-HT$_{2A}$ receptors in the IPN, are involved in the aversive properties associated with MRN serotonergic neuron activity. Further identification of the mechanisms underlying 5-HT$_{2A}$ signaling in the IPN will provide insights into a better understanding of the processing of rewards and aversive stimuli and the biological basis of mental disorders where aberrant processing of rewards and aversive

information are observed, such as in drug addiction and major depressive disorder.

## Methods

### Animals

All mice were handled in accordance with the ethical guidelines of the Kyoto University animal research committee (approval code: 13-41-2, 19-41-1,2,3), the animal research committee of Hokkaido University (approval code: 18-0070), and the animal research committee of Kanazawa University (approval code: AP-204167). Adult male C57BL/6J mice (7–11 weeks old; Japan SLC) were housed in groups (no more than six mice in an individual cage) in a plastic cage with wooden bedding and free access to food (MF, Oriental Yeast) and water, and kept under constant ambient temperature ($22 \pm 2\,°C$) and humidity ($55 \pm 10\%$), with 12 h light-dark cycles, unless otherwise stated. Mice were randomly assigned to each experimental group.

For the fixed-time schedule task, adult male C57BL/6N mice supplied from Japan SLC Co. Ltd. (aged >70 days) or adult male and female Tph2-tTA::tetO-ChR2(C128S)-eYFP bi-transgenic mice (8–20 weeks old) and their littermates were used. Approximately, 50% of the bi-transgenic mice were female. tetO-ChR2(C128S)-eYFP knock-in mice (RRID:IMSR_RBRC05454) and Tph2-tTA BAC transgenic mice (RRID:IMSR_RBRC05846) have been described in previous reports[39,40]. We previously demonstrated that Tph2-tTA:tetO-ChR2(C128S)-eYFP bi-transgenic mice selectively express ChR2 in almost all the serotonin neurons of the DRN and MRN, and that blue light application to the raphe nuclei increased the firing rates of serotonin neurons and yellow light application returned it to the baseline[41]. These mice were backcrossed to the C57BL/6N strain for more than ten generations. Animals were housed at $25 \pm 2\,°C$ and relative humidity of 40–50% in rooms where the light was turned on from 19:00 to 07:00 h. The tests were performed during the dark period (07:00 to 19:00 h). Mice were housed in groups of 3–5 until they received surgery and single-housed following surgery until the end of the experiment. Food and water were provided *ad libitum* until the beginning of the experiment.

### Production and purification of viral vectors

AAVs were prepared according to the previous report[15]. Lenti-X 293 T cells (Clontech) were grown to 60–70% confluency, and 8 μg of pHelper, 5 μg of pAAV-DJ Rep-Cap (Cell biolabs) and 5 μg of transfer plasmid were transfected with polyethylenimine (24765-1, Polysciences). After 60–72 h of incubation, the supernatant was aspirated, and 500 μL of 1× Gradient Buffer was added to the cells on each plate and then collected. The cell suspension was frozen in liquid nitrogen for 10 min and placed in a 55 °C water bath until the cells were completely thawed. This freeze–thaw cycle was repeated three more times. After the addition of 0.5 μL of benzonase (E1014-5KU, Sigma-Aldrich), the lysate was incubated at 37 °C for 60 min, centrifuged for 15 min at $3000 \times g$, and then the supernatant was collected. A discontinuous density gradient of 15%, 25%, 40%, and 58% iodixanol was prepared in an ultracentrifuge tube, and the supernatants were dripped onto the top layer of the density gradient. The tube was ultracentrifuged at 48,000 rpm (average $248,113 \times g$), 18 °C for 1 h in a 50.4Ti rotor (Beckman-Coulter). After ultracentrifugation, a syringe with an 18-gauge needle was inserted approximately 1–2 mm below the boundary between 40% and 58% gradient buffer layers, and about 0.5 mL of solution was slowly extracted. This was aliquoted and stored at −80 °C until use. The titer was measured by qPCR and estimated to be approximately $1.0 \times 10^{13}$ vg/mL.

### Stereotaxic surgery

Mice were anesthetized with isoflurane, and stereotaxic surgeries were conducted using a small animal stereotaxic frame (Narishige) according to the mouse brain atlas by Franklin and Paxinos[89]. The AAVs were microinjected at 1 μL into the MRN (anteroposterior (AP) −4.3 mm,

mediolateral (ML) 0 mm, dorsoventral (DV) +4.5 mm from bregma) or the DRN (AP −4.3 mm, ML 0 mm, DV +3.4 mm from bregma). We used AAV-DJ encoding Venus (Tph2-Venus)[34], a ChR2 variant, CheRiff[39], fused with eGFP (Tph2-CheRiff-eGFP), a light-activated proton pump, eArchT[37] fused with eYFP (Tph2-eArchT-eYFP), a calcium indicator, GCaMP6s[33] (Tph2-GCaMP6s), or axon-targeted GCaMP6, axon-GCaMP6s[52] (Tph2-axon-GCaMP6s) under the control of the Tph2 promotor. We confirmed that the efficiency of AAV bearing GCaMP6s and axon-GCaMP6s was similar to that of the AAV bearing Venus ((GCaMP6s) specificity $95.9 \pm 0.9\%$, coverage $92.0 \pm 2.3\%$; (axon-GCaMP6s) specificity $94.4 \pm 1.0\%$, coverage $90.7 \pm 2.7\%$). Owing to the membrane localization of CheRiff and eArchT, it was not possible to reliably measure the specificity and coverage of these AAVs.

For fiber photometry experiments and optogenetic experiments, 0–2 weeks after the viral injection, the animals were implanted with optic fiber (0.50 numerical aperture (NA), 250 μm/240 μm (clad/core); CK-10, Mitsubishi Rayon) so that the tip of the fiber was placed just above the MRN (AP −4.3 mm, ML 0 mm, DV +4.5 mm from bregma), IPN (AP −3.8 mm, ML 0 mm, DV +4.7 mm from bregma), vHP (AP −3.1 mm, ML 3.5 mm, DV +3.7 mm from bregma), MHb (AP −1.7 mm, ML 0.2 mm, DV +2.4 mm from bregma), LHb (AP −1.7 mm, ML 0.4 mm, DV +2.5 mm), PVT (AP −1.4 mm, ML 0 mm, DV +3.0 mm), or DRN (AP −4.3 mm, ML 0 mm, DV +3.4 mm from bregma). The fiber was implanted unilaterally for fiber photometry experiments, while the fibers in vHP and MHb for optogenetic manipulation were implanted bilaterally (vHP; AP −3.1 mm, ML ±3.5 mm, DV +3.7 mm from bregma, MHb; AP −1.7 mm, ML ±0.2 mm, DV +2.4 mm from bregma). An optical fiber was placed in a ceramic ferrule and inserted toward the target region. The ferrule was supported with a stainless-steel screw and dental acrylic. The mice were individually housed for at least 7 days to recover before behavioral analyses.

For the fixed-time schedule task, mice were anesthetized with isoflurane (4% for induction and 1% for maintenance) and fixed in a stereotaxic frame (Narishige). AAV-Tph2-CheRiff-eGFP or AAV-Tph2-Venus (400 nL) was injected through an injection cannula (35 gauge; Nanofil, NF35BL-2; World Precision Instrument) into the MRN (AP −4.4 mm, ML 1.8 mm, DV −4.3 mm at 20°). The virus solution was infused over 4 min at constant flow (100 nL/min) by a microinjection pump (Ultra Micro Pump II and Micro4 controller, World Precision Instruments), and the injector was left in place for 5 min after injection to allow diffusion. A week later, an optic fiber with a mirror tip at 45° (MA45; Doric Lenses) was implanted to target the MRN (AP −4.4 mm, ML −1 mm, DV −3.8 mm) and a chamber frame (CF-10, Narishige) was secured. Tph2-tTA::tetO-ChR2(C128S)-eYFP bi-transgenic mice received only optic fiber implantation and chamber frame surgery. After surgery, ointments containing antibiotics and steroids (Dolmycin, Zeria Pharmaceutical Co., Ltd.; Kenalog, Bristol-Myers Squibb) were applied to the wounds. A piece of jelly containing carprofen (MediGel, ClearH2O) was placed in the home cage, and the mice were housed individually and allowed to recover for seven days prior to behavioral experiments.

For electrophysiological experiments, mice were injected with AAV-Tph2-CheRiff-eGFP or AAV-Tph2-eArchT-eYFP at 500 nL into the MRN.

For Retrobeads injection experiments, the animals have injected Green Retrobeads IX (78G180, Lumafluor) at 100 nL into the IPN (AP −3.8 mm, ML 0 mm, DV +4.9 mm from bregma), vHP (AP −3.1 mm, ML 3.5 mm, DV +3.7 mm from bregma) or MHb (AP −1.7 mm, ML 0.2 mm, DV +2.4 mm from bregma).

For pharmacological experiments, 0–2 weeks after the viral injection, the animals were implanted with a guide cannula (CXG-5.1 (T), Eicom) so that the tip of the cannula was placed into the IPN (AP −3.8 mm, ML 0 mm, DV +4.9 mm from bregma). The cannula was supported with a stainless-steel screw and dental acrylic. A dummy cannula (CXD-5.1 (T), Eicom) was inserted into the guide cannula to

preserve the dosing channel until drug administration. The mice were individually housed for at least 7 days to recover.

## Histology and imaging

After behavioral analyses, all mice were anesthetized under sodium pentobarbital (Somnopentyl, Kyoritsu Seiyaku, 50 mg/kg) or a mixture of medetomidine (Domitor, Zenoaq, 0.3 mg/kg), midazolam (Dormicum, Maruishi Seiyaku, 4.0 mg/kg), and butorphanol (Betorfal, Meiji Seika Pharma, 5.0 mg/kg). Mice were transcardially perfused with phosphate-buffered saline (PBS) followed by 4% paraformaldehyde (02890-45, Nacalai Tesque) in phosphate buffer (PB). After perfusion fixation, the brains were removed, and equilibrated in 15% sucrose in PB for overnight. The brains were frozen and cryosectioned into 30 μm-thick coronal sections with the cryostat (Leica CM3050S; Leica Biosystems) and stored at −80 °C until immunohistochemical processing. For immunohistochemistry, the sections were immersed in 0.25% Triton X-100 (Nacalai Tesque) in PBS (PBS-T) for permeabilization and then incubated overnight at 4 °C with rabbit polyclonal anti-green fluorescent protein (GFP) antibody (1:2000; A-11122, Thermo Fisher Scientific), sheep polyclonal anti-tryptophan hydroxylase (Tph) antibody (1:200; AB1541, Merck Millipore), goat polyclonal anti-SERT antibody (1:200; HTT-Go-Af970, Nittobo Medical), mouse monoclonal anti-c-Fos antibody (2H2) (1:200; NBP2-50037, Novus Biologicals), rabbit polyclonal anti-5-HT$_{2A}$ receptor antibody (1:500; ASR-033, Alomone), or rabbit polyclonal anti-5-HT$_{2A}$ receptor antibody (1:100; #24288, ImmunoStar), followed by incubation with Alexa Fluor 594-labeled donkey anti-rabbit IgG (1:200; A-21207, Thermo Fisher Scientific), Alexa Fluor 647-labeled donkey anti-sheep IgG (1:200; A-21448, Thermo Fisher Scientific), Alexa Fluor 488-labeled donkey anti-rabbit IgG (1:200; A-21206, Thermo Fisher Scientific), Alexa Fluor 594-labeled donkey anti-goat IgG (1:200; A-11058, Thermo Fisher Scientific), Alexa Fluor 594-labeled donkey anti-sheep IgG (1:200; A-11016, Thermo Fisher Scientific), or Alexa Fluor 647-labeled donkey anti-mouse IgG (1:200; A-31571, Thermo Fisher Scientific) for 2 h at room temperature. The primary and secondary antibodies were dissolved in PBS-T with 5% horse serum and with 2% horse serum, respectively. The sections were then washed in PBS and mounted on glass with Fluoromount/Plus (K048, Diagnostic BioSystems) or DAPI Fluoromount-G (0100-20, Southern Biotech). Retrobeads was not immunostained, and the raw fluorescence was observed. Fluorescence was visualized using laser scanning confocal microscopy (FV10i, Olympus). AAV infection was verified immunohistochemically. The data points obtained from the mice with failed AAV infection, fiber implantation, or cannula implantation were excluded from the analysis. The numbers of the animals used were empirically determined and were similar to the previous reports[16,52,68,90,91].

For verification of AAV infection, fiber implantation, and cannula implantation, target brain regions (MRN, IPN, vHP, MHb, LHb, PVT, and DRN) were determined by anatomical landmarks with the guidance of the mouse brain atlas[89]. For the determination of AAV specificity and coverage, the number of Tph2-positive and AAV-positive cells in the MRN were counted in at least 3 sections within 150 μm anteroposterior to the slice containing injection site and were averaged, from −4.16 to −4.48 mm from the bregma. The cellular resolution images of coronal sections of interest were acquired through the 10× and 60× objective, using laser scanning confocal microscopy (FV10i, Olympus) and were processed with FV10i-SW software (Olympus) and ImageJ software (National Institutes of Health).

For verification of fiber placements in transgenic mice, mice were sacrificed under deep anesthesia (urethane, U2500, Sigma-Aldrich, 2 g/ kg, i.p.) after the completion of these experiments. Each brain was rapidly removed, cryoprotected with Tissue-Tek® O.C.T Compound (Sakura® Finetek USA, Inc.), and frozen at −80 °C. Coronal sections (40-μm thick) were cut on a cryostat and thaw-mounted onto slides. Optical fiber placements were verified under a confocal laser-scanning

microscope FV1000 (Olympus) according to the atlas[89]. Data from mice with incorrect placements were excluded from the analysis.

Immunohistochemistry was performed as described previously[92] with minor changes. In brief, mice were anesthetized with urethane (U2500, Sigma-Aldrich, 2 g/kg, i.p.). Following intracardial perfusion with 4% paraformaldehyde in PBS (pH 7.2), mice brains were post-fixed overnight, placed in 0.1 M PB containing 20% sucrose, and sectioned at a thickness of 40 μm. Sections were incubated successively with 5% normal donkey serum for 20 min and a mixture of primary antibodies overnight. The primary antibodies used were mouse anti-GFP (1:1000; 012-20461, WAKO) and rabbit anti-tryptophan hydroxylase 2 (1:10,000; Nakamura et al., 2008)[93]. Sections were washed three times in PBS-T before the next incubation using secondary antibodies. The secondary antibodies used were donkey anti-mouse Alexa 488 (A-21202, Invitrogen) and donkey anti-rabbit (Cy3; 711-165-152, Jackson ImmunoResearch). Images were captured with a confocal laser-scanning microscope FV1000 (Olympus) for high magnification (10× and 40×) and a fluorescence microscope BX50 (Olympus) for low magnification (4×).

## Fiber photometry

To record fluorescence signals, we used a custom-built fiber photometry system fabricated according to the previous reports[94,95] with minor changes. Blue light emitted from a 470-nm LED (M470L3 or M470L4, Thorlabs) was reflected by a dichroic mirror (MD498, Thorlabs) and focused by a 10× objective lens (MRH00101, Nikon) to the tip of the fiber patch cable (M123L01 and M73L01, Thorlabs) whose another tip was connected to the implanted optical fiber. The LED power was adjusted at the tip of the optical fiber to 0.02–0.03 mW to minimize bleaching. The fluorescence was bandpass filtered (FBH520-40, Thorlabs) and focused by a tube lens (58-520, Nikon) to a CMOS camera (CS2100M-USB, Thorlabs) regulated by the ImageJ software plugin Micro-Manager[96]. The fluorescence images were captured at 10 Hz.

For photometry data analysis, regions of interest were manually drawn around the fiber on the basis of a detected image, then the average fluorescence intensity was calculated for each image. We also draw regions of interest out of the fiber on the basis of a detected image and calculated the average intensity to account for extraneous, non-fluorescence-related light potentially contributing to the signal. We subtracted this acquired offset from the fluorescence intensity for each image. We derived the values of fluorescence change ($\Delta F/F_0$) by calculating $(F - F_0)/F_0$, namely by subtracting the baseline fluorescence ($F_0$) from the fiber fluorescence at each time point ($F$) and dividing that value by the $F_0$, where $F_0$ was set during the 1 s before the trigger events. For fiber photometry data analysis for DRN serotonin neurons, $F_0$ was the baseline fluorescence signal averaged over a 1.5-s-long control time window, which was typically set at 0.5 s preceding the trigger events[16].

## Slice preparation and electrophysiological recordings

Three weeks after the viral injection, mice were anesthetized with isoflurane and transcardially perfused with ice-cold N-methyl-D-glucamine (NMDG)-based solution containing (in mM): NMDG, 92; D-glucose, 25; N-2-hydroxyethylpiperazine-N′-2-ethanesulfonic acid (HEPES), 20; NaHCO$_3$, 26; CaCl$_2$, 0.5; MgSO$_4$, 10; KCl, 2.5; NaH$_2$PO$_4$·2H$_2$O, 1.25; Thiourea, 2; Sodium ascorbate, 5; Sodium pyruvate, 3; N-Acetyl-L-cysteine, 12 (oxygenated with 95% O$_2$ and 5% CO$_2$, pH 7.4 with hydrochloric acid). The brain was removed, and coronal slices (250 μm thick), including the MRN, were cut with a microslicer (VT1200S; Leica) and incubated at 32–34 °C for 10 min and then kept at room temperature for 5 min in a chamber containing NMDG-based solution. After that, slices were stored with artificial cerebrospinal fluid (aCSF) containing (in mM): NaCl, 124; KCl, 3; NaHCO$_3$, 26; NaH$_2$PO$_4$, 1; CaCl$_2$, 2.4; MgCl$_2$, 1.2; D-glucose, 10; and bubbled with 95% O$_2$/5% CO$_2$

(pH 7.4) at room temperature. The slices were mounted in a recording chamber on a fluorescence microscope (BX-51WI; Olympus) equipped with an infrared camera (IR-1000; Dage-MTI), and continuously perfused with aCSF at a flow rate of 2.0–2.5 mL/min.

Data were amplified using a MultiClamp 700B amplifier and stored on a computer using the pClamp10 software version 10.7 (Molecular Devices). Individual neurons were visualized with the fluorescence microscope. Whole-cell current-clamp recordings were obtained from eGFP- or eYFP-positive neurons in the MRN by patch pipettes under visual control. Pipettes were prepared from borosilicate glass capillaries and filled with an internal solution containing (in mM): K-gluconate, 140; KCl, 5.0; HEPES, 10; MgCl$_2$, 2.0; Na$_2$ATP, 2.0; Na$_3$GTP, 0.3; EGTA, 0.2; (pH 7.3 with KOH). The electrode resistance was 4–7 MΩ in the aCSF. All recordings were performed at 32–34 °C. For recording action potentials induced by CheRiff stimulation in eGFP-labeled neurons, blue LED light (470 nm, 5.92 mW, 20 Hz frequency, 10 ms duration, 40 pulses) was applied. For confirming the suppressive effect of eArchT stimulation on action potentials in eYFP-labeled neurons, a depolarizing current (0–30 pA, 10 s duration) was injected to evoke action potentials with approximately 1–2 Hz, and green LED light (530 nm, 1.58 mW, 3 s duration, 3.5 s after the start of depolarizing current injection) was applied.

### RT-PCR

Total RNA was isolated from the IPN tissue block punched out from the coronal mouse brain section using NucleoSpin RNA kit (740955, Takara Bio) according to the manufacturer's instruction. Purified total RNA (150 ng) was reverse-transcribed using RevertraAce qPCR RT Master Mix (FSQ-201, Toyobo). Diluted cDNA was PCR amplified using Luna Universal qPCR Master Mix (N3003, New England BioLabs) with primers. Amplified DNA was electrophoresed in agarose gel and visualized using a UV transilluminator. Sequences of primers were following; Htr2a-Fw 5′-aaaagcatgcaaggtgctgg-3′, Htr2a-Rv 5′-ttctttgcagatgacggcca-3′, Htr2b-Fw 5′-ttcaggccaatcagtgcaac-3′, Htr2b-Rv 5′-ggatggcgatgcctattgaa-3′, Gapdh-Fw 5′-tgtccgtcgtggatctgac-3′, Gapdh-Rv 5′-cctgcttcaccaccttcttg-3′.

### Optogenetic manipulation

The optic fibers implanted in mice were connected to the optical fiber patch cables. Light emitted from the diode-pumped solid-state (DPSS) laser (Beijing Viasho Technology) was converged to the fiber optic by the FC/PC collimator, which was connected to the optical fiber patch cable. The DPSS laser was driven by the microcontroller (Arduino Uno, Arduino SRL) to generate the pulse width modulation signal. For the Tph2-eArchT and Tph2-Venus mice, green light illumination was delivered throughout the conditioning session of the CPP test (Figs. 1 and 6; 532 nm, 5 mW at the tip of the fiber, continuous, 20-s ON, 10-s OFF, for soma stimulation and axon terminal stimulation). In the self-stimulation test, brief green light illumination was delivered just after nose-poking (Fig. 1; 532 nm, 5 mW at the tip of the fiber, 3-s continuous, for soma stimulation). For the Tph2-CheRiff and Tph2-Venus mice, blue light illumination was delivered throughout the conditioning session of the CPA test (Figs. 2, 7, and 8; 473 nm, 5 mW at the tip of the fiber, 20 Hz frequency, 10-ms ON and 40-ms OFF (20% duty cycle), for soma stimulation and axon terminal stimulation) and during sucrose solution intake from a light stimulation-paired bottle (Fig. 3 and Supplementary Fig. 5; 473 nm, 5 mW at the tip of the fiber, 20 Hz frequency, 10-ms ON, 40-ms OFF, for soma stimulation). In Supplementary Fig. 4 and 6, to evaluate c-Fos expression, blue light illumination was delivered 20 min in a chamber of the same size as the chamber of the CPA test (473 nm, 5 mW at the tip of the fiber, 20 Hz frequency, 10-ms ON, 40-ms OFF, for soma stimulation).

For MRN manipulation during the fixed-time schedule task in AAV-injected mice, blue light (473 nm, 10 mW at the tip of the implanted optic fiber) application to the MRN was generated by a laser light source (COME2-LB473/589/100; Lucir). The light delivery was controlled via TTL pulses driven by a microcontroller board (Arduino Uno, Arduino SRL). Mice that acquired an anticipatory number of licks/seconds superior to 1.5 were tested the next day. Based on the stable and stereotypic pattern of licking behavior that developed in this task, the anticipatory licking phase was determined as starting from −6 s and ending at 0 s, the latter corresponding to the reward delivery. A test session consisted of 3 phases of 10 min, identical to the training runs. The first phase, Pre was used as a baseline of the licking behavior developed during the training phase. During the following phase Stim, a 2-s long blue light pulse (473 nm, 10 mW, 20 Hz, 10-ms ON, 40-ms OFF) was delivered from the onset of each reward delivery. Finally, phase Post was a replicate of phase Pre to assess a potential long-lasting effect of the optogenetic manipulation.

For MRN manipulation during the fixed-time schedule task and CPA test in bi-transgenic mice, blue light (473 nm, 1–3 mW at the tip of the implanted optic fiber) or yellow light (575 nm, 1–4 mW at the tip of the implanted optic fiber) application to the MRN was generated by a LED light source (SPECTRA 2-LCR-XA light engine; Lumencor). A test session consisted of 3 phases of 10 min, identical to the training runs. The first phase, Pre was used as a baseline of the licking behavior developed during the training phase, during which a 0.5-s constant yellow pulse was delivered from the onset of each reward delivery and followed 1.5 s later by a second 0.5-s constant yellow pulse. During the following phase Stim, the same light procedure was used, but the first pulse was blue. Finally, phase Post was a replicate of phase Pre to assess a potential long-lasting effect of the optogenetic manipulation.

### Behavioral analyses

**Sucrose and quinine solution licking test.** The mice were water-deprived for 13–18 h per day for three days before the test. Before the test was conducted, the mice were placed for 5–8 h per day in a test cage (dimensions: 20.8 × 13.6 × 11.5 cm, Length × Width × Height) and allowed to drink *ad libitum* 10% (w/v) sucrose solution for two consecutive days to acclimate to a test cage. The next day, the mice attached with a fiber cable were placed in a test cage equipped with a bottle filled with 10% sucrose solution. The test and the fiber photometry recording were performed for 10 min in a free-moving setup. The mice had free access to the solution during the test. The timing of the lick onset was curated through analyses of recorded movies. For comparison of fluorescence response induced by different concentrations of sucrose, the mice were water-deprived, as mentioned above, for two consecutive days. Then, the mice attached with a fiber cable were placed in a test cage and were presented with 10% and 0.5% sucrose solution alternately. For the recording of the response to water and quinine solution, the mice were water-deprived, as mentioned above, for two consecutive days. Then, the mice attached with a fiber cable were placed in a test cage and were presented with a water solution. After the water solution was licking, the bottle filled with 5 mM quinine solution (quinine hydrochloride, CAS: 130-89-2; Tocris) was replaced with a bottle of water. Fiber photometry recording was performed throughout the session.

**CPP test.** The CPP test was performed according to previous reports[97,98] with minor changes. The CPP apparatus consisted of two equal-sized compartments (dimensions: 15 × 24 × 30 cm, Length × Width × Height) with distinct tactile and visual cues. One compartment was white with a textured floor, and the other one was black with a smooth floor. In pretest sessions on Day 1, mice without the fiber patch cable were allowed to explore two compartments freely for 900 s, and the time spent in each compartment during the exploration period was measured using ANY-maze tracking software (ANY-maze version 6.0, Stoelting). Mice that spent more than 80% (>720 s) of the total time in one compartment in the pretest or that showed a difference of >200 s in time spent in one compartment between the pretest were excluded

from the following procedures (No mice were excluded by this criterion throughout the study). We used a bias-like protocol[36] and designated the compartment in which each mouse spent less time (<450 s) in the pretest as the light stimulation-paired compartment for that animal. In conditioning sessions on Days 2 and 3, mice were connected to a fiber optic patch cable. During the first half of the day's conditioning, one group of mice was confined to the stimulation-paired compartment for 20 min with light delivery, and the other group of mice was confined to the non-stimulation-paired compartment for 20 min without light delivery. During the second half of the conditioning, mice that received light delivery in the first half were confined to the non-stimulation-paired compartment for 20 min without light delivery. The other group of mice was confined in the stimulation-paired compartment for 20 min with light delivery. The mice were randomly grouped and conditioned. An interval of at least 4 h was left between the first and second half of the conditioning. In posttest sessions on Day 4, mice without the fiber patch cable were allowed to freely explore the two compartments for 900 s, and the time spent in each compartment during the exploration period was measured. The CPP scores were calculated by subtracting the time spent in the light-paired compartment during the pretest from that during the posttest.

**Nose poke self-stimulation in an operant chamber.** The operant conditioning chamber (dimensions: 15.24 × 13.34 × 12.7 cm, Length × Width × Height; Med Associates) was equipped with a nose poke port (ENV-303M; Med Associates) and encased in a sound-attenuating box. Nose poking through the hole resulted in the delivery of green light stimulation (532 nm, 1 pulse, 3 s) into the target region through the optical fiber. Light stimulation was delivered for each nose-poking response. The animals were plugged into the fiber-optic patch cord and placed into the chamber, and subsequently allowed to self-stimulate the target nucleus for 30 min. The mice were allowed to perform the task for four consecutive days, and the number of nose pokes was counted for each day.

**Tail pinch test.** The mice attached with a fiber-optic patch cord were placed in the test cage (dimensions: 20.8 × 13.6 × 11.5 cm, Length × Width × Height) for at least 5 min for habituation before the test. Then, the experimenter gave the mice a tail pinch as an aversive stimulus. For all mice, the same experimenter performed tail pinches using a bulldog serrefine clamp (18050-50, Fine Science Tools). The pinch continued not longer than 20 s to minimize the animal's pain, and the fluorescent response before and after the pinch was recorded. The timing of the pinch was curated through analyses of recorded movies.

**Cue-induced fear conditioning.** Auditory cue-induced fear conditioning was performed according to the previous report[38]. On day 1, mice without fiber patch cords were placed in the operant chamber (15.24 × 13.34 × 12.7 cm, Length × Width × Height) equipped with stainless steel grid rod floor and speaker for generation of auditory cue (2900 Hz; ENV-323AM, Med Associates) for 180 s. Then, an auditory cue (30 s duration, 2900 Hz) was applied for 30 s. Foot shock (0.6 mA, 2 s duration; ENV-414S, Med Associates) was applied after initiation of the auditory cue and terminated simultaneously with the auditory cue. This cycle was repeated twice with 30-s intervals (3 shocks in total). After conditioning, the mice were returned to their home cage. On day 2, mice with fiber patch cords were placed in the chamber with different floor textures (paper towels), different odors (vanilla oil), and different illumination colors for 180 s. Afterward, an auditory cue (2900 Hz) was applied, and GCaMP fluorescence in the MRN was measured.

**CPA test.** The same apparatus was used as the CPP test. After the same pretest (900 s, no mice were excluded by the abovementioned

criterion throughout the study) as the CPP test, we designated the compartment in which each mouse spent more time (>450 s) in the pretest as the light stimulation-paired compartment for that animal. In conditioning sessions on Days 2 and 3, mice were connected to a fiber optic patch cable. During the first half of the day's conditioning, one group of mice was confined to the stimulation-paired compartment for 20 min with light delivery, and the other group of mice was confined to the non-stimulation-paired compartment for 20 min without light delivery. During the second half of the conditioning, mice that received light delivery in the first half were confined to the non-stimulation-paired compartment for 20 min without light delivery. The other group of mice was confined in the stimulation-paired compartment for 20 min with light delivery. The mice were randomly grouped and conditioned. An interval of at least 4 h was left between the first and second half of the conditioning. In posttest sessions on Day 4, as in the CPP test, mice without the fiber patch cable were allowed to freely explore the two compartments for 900 s, and the time spent in each compartment during the exploration period was measured. The CPA scores were calculated by subtracting the time spent in the light-paired compartment during the pretest from that during the posttest.

For the CPA test with drug administration, the mice were injected intraperitoneally with vehicle (1% DMSO in saline), WAY-100635 (0.5 mg/kg, 5-HT$_{1A}$ receptor antagonist, CAS: 634908-75-1; Abcam), or MDL-100907 (0.5 mg/kg, 5-HT$_{2A}$ receptor antagonist, CAS: 139290-65-6; R&D Systems) 30 min prior to 20-min conditioning[99–101]. In the experiments with microinjection, the mice were injected into the IPN with vehicle (1% DMSO in saline, 0.2 μL), WAY-100635 (1 μg in 0.2 μL), or MDL-100907 (0.1 μg in 0.2 μL) 5 min prior to 20-min conditioning[58,102–105]. In the experiments with microinjection of 5-HT$_{2A}$ agonist, the mice were injected into the IPN with vehicle (saline, 0.5 μL) or TCB-2 (0.05 μg in 0.5 μL, 5-HT$_{2A}$ receptor agonist, CAS: 912342-28-0; Tocris) 5 min prior to 20-min conditioning.

**Two-bottle choice test.** The mice were water-deprived for 13–18 h per day before the conditioning before each session. In the conditioning session, the mice were placed for 5–8 h per day in a test cage (dimensions: 20.8 × 13.6 × 11.5 cm, Length × Width × Height) and were allowed to take 10% sucrose solution in two identical bottles for 30 min × two times per session a day interleaved with at least 4 h interval. The amount of solution intake (g) in each bottle was recorded in each conditioning session. In the test session, the mice attached to a fiber optic patch cable were allowed to take 10% sucrose solution in two identical bottles. One bottle was associated with light stimulation (ON bottle), and another bottle was not (OFF bottle). Light stimulation was given whenever the mice licked the ON bottle. The position of ON and OFF bottles was counterbalanced between the first and second test sessions. The amount of solution intake (g) in each bottle was recorded per test day. We summed the solution intake in the first and second test sessions for each bottle to minimize possible differences due to bottle preference and left–right bottle position.

**Fixed-time schedule task.** After recovery from surgery, mice were water-deprived in their home cage until the end of the experiment. Mice were trained in a square behavioral chamber that was closed by an opaque black curtain. Each animal was kept in a covered elevated platform (3D printed), with its head fixed by two stabilized clamps holding the side of the chamber frame[106]. The heights of the tunnel and clamps were adjusted before each session to ensure comfort. A steel blunt-type needle was placed in front of the mouse's mouth and in front of an infrared beam-break sensor that was used to detect licks. Mice were allowed to voluntarily lick the spout. A 2 μL drop from a 10% sucrose solution was delivered through the needle every 10 s by a peristaltic pump (lab-hacks.com; no more available, but available at https://www.tindie.com/products/mindgame/high-precision-peristaltic-pump-hat/), controlled by a microcontroller board (Arduino Uno, Arduino SRL)

connected to a PC via a USB cable. A custom-written python script was used to collect the timing and number of licks and saved the data in a CSV-type file. Each daily session lasted for 30 min. Mice were allowed access to water in their home cage for 30 min following the end of a daily session.

**Surgery for oral infusion of sucrose solution.** Tph2-tTA::tetO-ChR2(C128S)-eYFP bigenic mice were anesthetized by intraperitoneal injection of a mixture of medetomidine, midazolam, and butorphanol[107]. We implanted the optic fiber to the MRN as described above and intraoral cheek fistula to the left cheek following a previous rat study[108], but with a few changes for mice. We used soft poly-urethane tubing (0.6 mm inner diameter and 1.0 mm outer diameter; Micro-Renathane, Braintree Scientific, Inc.) connected to a 22 gauge stainless steel needle. A customized Teflon washer (3 mm diameter) was glued to the other end of the tubing.

**Oral infusion of sucrose solution and taste reactivity analysis.** Water-deprived mice were fixed as described above. A 5 μL drop from a 20% sucrose solution was delivered through the intra-oral tube by a pump (OPR-7200, O'Hara & Co., Ltd.), controlled by a microcontroller board (Arduino Uno, Arduino SRL) connected to a PC via a USB cable. To check whether the apparatus works well and obtain prototypical face data (see Facial expression analysis), we infused the sucrose solution first. Following that, mice received three phases (Pre, Stim, and Post) at 10 min intervals. In the Stim phase, blue light was applied to the MRN for 500 ms at the timing of delivery of the sucrose solution. In this task, we did not apply yellow light to intentionally stop opto-genetic activation because the C128S mutant spontaneously deacti-vates within 1–2 min[109] which is far longer than 10 min intervals. We recorded the taste reactivity through the video camera (iVIS HF R52, Canon) at 30 fps. We counted the number of tongue protrusions from the mouth as taste reactivity and hedonic response to sucrose solution[42]. As a simpler measure of taste reactivity, we also counted the number of tongue appearances whenever the tongue was visible for 10 s after the delivery of the sucrose solution.

**Facial expression analysis.** We acquired facial video recordings at 30 fps using a USB 3.0 monochrome camera (BFS-U3-13Y3M-C, FLIR Integrated Imaging Solutions Japan Co., Ltd.), positioned on the right side of the mouse's head. Illumination was provided by a 940 nm IR LED array (K-Light, Cony Electronics Service Co., Ltd.). Before starting the experiments, the prototypical face was recorded before and after the oral infusion of sucrose solution. The acquired images were ana-lyzed following a previously described procedure[43]. Briefly, each frame was converted into a histogram of oriented gradients (HOG) vector. The average similarity to the prototypical face of sucrose-induced facial expression was calculated using the HOG before and after intra-oral sucrose infusion. We calculated the similarity index to compare the facial expression changes among three phases (Pre, Stim, and Post) as follows:

$$\text{Similarity index} = \frac{\text{after} - \text{before sucrose infusion}}{\text{after} + \text{before sucrose infusion}}$$

**Quantification of c-Fos positive cells**
The mice used for c-Fos quantification were not used in any other experiments. As described above, we conducted blue laser stimulation on the mice. Ninety min after the beginning of light stimulation, the mice were sacrificed, and the brains were harvested. Cryosections were prepared and processed as described in Histology and imaging sec-tion. The number of c-Fos positive cells was quantified in at least three different sections containing the MRN, IPN, vHP, or MHb, and averaged across sections.

**Drug administration**
For the CPA test with pharmacological intervention, the vehicle or drugs were administered intraperitoneally 30 min before the conditioning. For the CPA test with microinjection, the mice were implanted with a microinjection guide cannula (CXG-5.1 (T), Eicom) into the IPN. Five min before the conditioning, the vehicles or drugs were injected into the IPN (AP −3.8 mm, ML 0 mm, DV +4.9 mm from bregma) using a micro-syringe (7002KH, Hamilton) with an injection cannula (CXMI-5.4 (T), Eicom) attached to a polyethylene tube. The solution (0.2 μL) was infused at a rate of 0.2 μL min$^{-1}$ using a syringe pump (KDS 100, KD Scientific). The injection cannula was left in place for three additional min and then slowly withdrawn, and the dummy cannula was attached.

**Statistics and reproducibility**
For histochemical analyses, similar results were obtained in at least 10 (Supplementary Fig. 5c), 6 (Supplementary Fig. 5d), 5 (Supplementary Fig. 5e), and 3 (Fig. 5) mice. For histochemical analyses of 5-HT$_{2A}$ receptors (Supplementary Fig. 11g, h), experiments were performed once, but a similar staining pattern was commonly obtained by different antibodies both targeting 5-HT$_{2A}$ receptors as shown in Supplementary Fig. 11g, h. For RT-PCR analyses of Htr2a and Gapdh in IPN cDNA, experiments were performed once, but similar results were obtained in two biological replicates, as shown in Supplementary Fig. 11i.

**Data analysis**
The data are presented as mean ± s.e.m. Statistical significances were calculated using R (version 4.0.3), GraphPad Prism (versions 8 and 9, GraphPad Software, Dotmatics), and SPSS (version 23.0). Differences between two individual groups were compared using two-tailed unpaired Student's $t$-tests. Differences between two-time points of the same animal were compared using two-tailed paired Student's $t$-tests. For photometry, data were compared using 1000 permutations for an α-level of 0.05 to compare the values of $\Delta F/F_0$ at each time point with baseline values[16]. For group comparisons, data were compared using one-way or two-way analyses of variance (ANOVA), followed by Tukey *post hoc* tests or two-way repeated-measures ANOVA followed by Sidak *post hoc* tests. Licking behaviors in the fixed-time schedule task were analyzed using a three-factor mixed-design ANOVA with the phase and time as the within-subject factors and the virus type as the between-subject factor or two-factor repeated-measures ANOVA with the phase and time as the within-subject factors in transgenic mice. In all cases, $P$ values less than 0.05 ($P < 0.05$) were considered statistically significant. The variance within each group was analyzed by $F$-tests or Bartlett tests. The variance is similar between groups in all figures except Fig. 3d, Supplementary Figs. 4t, 7e, l, q, r, and 9h. Thus, Fig. 3d, Supplementary Figs 4t, 7e, l, q, r, and 9h were analyzed by two-tailed unpaired $t$-tests with Welch's correction.

**Reporting summary**
Further information on research design is available in the Nature Portfolio Reporting Summary linked to this article.

## Data availability
The source data are provided as a Source Data file. Other data that support the findings of this study are available from the corresponding authors on request. Source data are provided with this paper.

## Code availability
Custom code generated in this study is available at: https://github.com/YuYuB/MRN-Head-Fixed.

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

## Acknowledgements

We would like to thank Dr. Hiroyuki Hasegawa for kindly gifting us anti-tryptophan hydroxylase 2 antibodies. We would like to thank Drs. Akihiro Yamanaka and Kenji F. Tanaka for kindly providing us with Tph2-tTA and TetO-ChR2(C128S) mice. We would like to thank Dr. Hitomi Sasamori for her help in the breeding of transgenic mice. We would like to thank Drs. Mike Robinson and Kent Berridge for telling us how to conduct the surgery of oral infusion. We would like to thank Hitoshi Saito for his help in programming. We would like to thank Dr. Masaaki Sato for kindly lending us an infusion pump. We would like to thank Drs. Katsuyuki Kaneda and Satoshi Deyama for help on the CPP/CPA tests. We would like to thank Drs. Atsushi Miyawaki (pCSII-EF-Venus), Adam Cohen (CheRiff; addgene #51694), Karl Deisseroth (eArchT; addgene #35513), Douglas Kim and GENIE Project (GCaMP6s; addgene #40753), and Lin Tian (axon-GCaMP6s; addgene #112008) for providing us the constructs. This work was supported by Grants-in-Aid for Scientific Research from JSPS (to K.Nagayasu (JP20H04774, JP20K07064), to S.K. (JP18H04616, JP20H00491), to Y.O. (21K07473), to M.Y. (21H02668), to M.Kondo (JP22K11498), Grant-in-Aid for Nagai Memorial Research Scholarship from the Pharmaceutical Society of Japan (to H.K. (N-184403)), Grants-in-Aid for JSPS Fellows (to H.K. (JP20J12341), Y.N. (JP21J14215), and C.A. (JP21J21091)), AMED (to S.K. (JP20ak0101088h0003, JP21ak0101153h0001), to M.Kondo (JP21wm0525026, JP20lm0203007)), Smoking Research Foundation (to Y.O.), The Shimizu Foundation for Immunology and Neuroscience Grant (to K.Nagayasu), The Uehara Memorial Foundation (to K.Nagayasu), The Lotte Foundation (to K.Nagayasu), Takeda Science Foundation (to M.Kondo), SENSHIN Medical Research Foundation (M.Kondo).

## Author contributions

H.K., Y.B., K.Nagayasu, Y.O., and S.K. designed the study. H.K. and Y.B. contributed equally to this work. H.K. performed fiber photometry experiments, optogenetic experiments, pharmacological experiments, behavioral experiments, AAV production, histochemical analyses, and data analyses with the help of M.Koda, H.M., C.A., M.H., H.S., and M.Kondo. Y.B., K.T., and Y.O. jointly constructed the head-fixed behavioral setup, and Y.B. and Y.O. performed an optogenetic intervention on licking behaviors and histochemical analyses. N.N., K.Niitani, S.I., and K.K. performed electrophysiological experiments. N.N., Y.N., M.Koda, C.A., and K.Nagayasu constructed and produced AAVs. H.K. wrote the paper draft. Y.B., K.Nagayasu, Y.O., and S.K. jointly edited the paper. K.Nagayasu, Y.O., M.Y., and S.K. jointly supervised the project.

## Competing interests

The authors declare no competing interests.
