## [Peer Review File · Nature Communications]

Median raphe serotonergic neurons projecting to the interpeduncular nucleus control preference and aversionReviewers' Comments:

Reviewer #1:

Remarks to the Author:

Comments on NCOMMS-21-44544-T

The brain serotonin system modulates numerous behavioral processes. Serotonin neurons in the dorsal raphe nucleus attract much attention for their roles in reward processing. Although the median raphe nucleus (MRN) is also a major source of serotonin to the forebrain, it has remained elusive whether and how MRN serotonin neurons contribute to reward processing. Previous studies on the MRN often indicate a role of this nucleus and its serotonin neurons in modulating theta rhythm, locomotion, and memory formation (Graeff et al., 1980; Vertes et al., 1994; Kusljic et al., 2003; Kocsis et al., 2006; Fernandez et al., 2017). Here, Kawai et al. examined the potential functions and circuit mechanisms of MRN serotonin neurons in reward processing. Fiber photometry of Ca²⁺ signals indicate that these neurons are inhibited by reward (sucrose) consumption and activated by aversive stimuli (tail pinch). Inhibiting these neurons promotes reward-like behavior, whereas activating the cells produces behavioral aversion. Finally, the authors showed that the projection from the MRN serotonin neurons to the interpeduncular nucleus (IPN) seems particularly important for the behavioral effects. The authors conclude that the serotonergic projections from the MRN, particular the one to the IPN, negatively regulate reward processing.

These results could potentially expand our understanding of the behavioral functions of brain serotonergic system, and may explain how SSRI drugs, while increasing brain serotonin quite rapidly, do not produce strong pleasure in humans. I have the major concern on reconciling the well-established functions of the MRN neurons in hippocampal theta, locomotion, and memory formation with the authors' observations and conclusions. Even the IPN, which forms strong connections with the MRN and the nucleus incertus, is closely involved in modulating hippocampal theta, animal locomotion, and memory processes. The authors did not monitor animal locomotor activity and hippocampal theta during the fiber photometry experiments, so it remains unclear how much changes in Ca²⁺ signals were related to the valence of the stimuli or animal locomotion. The authors did report that activating or inhibiting MRN serotonin neurons did not have a significant effect on overall locomotion (distance travelled). However, the causality could be opposite in direction: locomotor activity changes the neuronal activity of MRN serotonin neuron but not the other way around. Indeed, some of the behavioral effects shown in this study might be interpreted using prior observations that MRN serotonin neurons suppress hippocampal theta. To more convincingly support their conclusions, the authors should simultaneously record neuronal Ca²⁺ signals, hippocampal theta, and animal locomotion in real time. In addition, they should test multiple rewarding and aversive stimuli, each of which has distinct relationship to locomotion. For example, one reward stimulus is associated with locomotion (appetitive chasing), and another with no locomotion (sucrose consumption as shown here). By the same token, one aversive stimulus is associated with locomotion (foot shock), and another with no locomotion (freezing to a pain-predicting neutral cue). This new set of recordings might help unravel how MRN serotonin neurons encode stimulus valence and animal behaving states.

I have other more specific concerns:

1. Figure 4 shows strong terminal labeling in the VTA and the PVT, as well as some clear labeling in the LHb. All of these areas are involved in reward processing. The authors instead focused on the ventral hippocampus, IPN, and MHb to examine the potential downstream targets of MRN serotonin neurons. Please explain the rationale behind the choices and how the VTA and PVT might not be important despite the dense innervations.
2. In addition, the IPN is adjacent to the MRN, and the serotonergic fibers from the MRN likely pass through the IPN en route to forebrain centers through the fasciculus retroflexus. This may explain how stimulating or inhibiting the IPN produces the strongest effect. This spatial closeness may even confound data interpretation for the fiber photometry and retrograde labeling experiments.
3. The authors attempted to confirm the validity of optogenetic stimulation by showing more c-Fos⁺ cells in the IPN of CheRiff-stimulated mice (Extended Data Fig. 5). This approach is problematic, since the authors showed that stimulating MRN serotonin neurons did not increase c-

Fos expression in the vHP but manipulating the terminals of these neurons in the vHP changed the reward and aversion processing. Therefore, c-Fos is not a very good marker of neuronal activation here. Importantly, the authors did not provide any good evidences that their choices of parameters for optogenetic interventions effectively changed neuronal activity in vivo or ex vivo (brain slices). I am particularly concerned about the choice of eArchT, which in our hands may not be very effective in suppressing neuronal firing in some brain areas but at the same time may induce post-inhibition rebound.

4. The authors showed that intra-IPN administration of MDL-100907 reversed the reduction in the CPA score induced by MRN activation (Figure 7). The authors need to validate 5-HT_{2A} expression by immunostaining. Due to the close distance between the IPN and the VTA, MDL-100907 might diffuse to the VTA and thus blocked the 5-HT_{2A} receptors there. Moreover, there are multiple serotonin receptors that are richly expressed in the IPN, either in the axonal terminals of MHB neurons (e.g. 5-HT₄) or in IPN neurons. The authors need to take great caution when making the statement that "5-HT_{2A} receptors in the IPN mediate the aversive properties associated with MRN serotonergic neuron activity". Nowadays pharmacological evidences are often backed up by in vivo genetic manipulations, such as Crispr-Cas9-mediated knockout of 5-HT_{2A} in the IPN here, to reach a more clear-cut conclusion.

Minor concerns:

1. The efficiencies of viral expression are critical for the reliabilities of experiments. We recommend the authors to verify the expression efficiencies of every virus batch and check the rates of transgenes expression in serotonin neurons.
2. In Fig 1h, the GCaMP signals increase before the sucrose licking. Does this mean that reward anticipation increases the activities of MRN serotonin neurons?
3. The title of Extended Data Fig.5 "c-Fos expression in the MRN was induced by light stimulation in IPN, vHP, and MHB", is consistent with the description in the main text.

Reviewer #2:

Remarks to the Author:

In this study, Kawai and colleagues show the role of MRN in processing reward and aversive information. They find that

1. MRN serotonergic neurons increase and decrease their activity by an aversive stimulus tail pinch and an appetitive stimulus sucrose solution, respectively.
2. Optogenetic excitation and inhibition of MRN serotonergic neurons induce aversive effect measured by the conditioned place aversion (CPA) test and reward effect measured by the conditioned place preference (CPP) test, respectively.
3. MRN serotonergic neurons projecting to the IPN and the vHP increase and decrease their activity by the tail pinch and sucrose solution consumption, respectively.
4. Optogenetic excitation and inhibition of MRN serotonergic neurons projecting to the IPN and the vHP induce aversive effect measured by CPA test and reward effect measured by the CPP test, respectively.
5. 5-HT_{2A} receptors in the IPN are necessary in the processing of aversive stimuli.

These results show the precise mechanism how MRN serotonergic neurons work in processing reward and aversive stimuli. The finding that MRN serotonergic neurons are activated by aversive stimuli is contrast to DRN serotonergic neurons which are mainly activated by reward stimuli. This study provides very important data. I have a few issues to be clarified for aversive processing of MRN serotonergic neurons.

1. I am very interested in whether MRN serotonergic neurons respond to other aversive stimuli other than the tail pinch. The tail pinch induces pain. Does the CPA score decrease because optogenetic activation of MRN serotonergic neurons causes pain? For example, when quinine

solution, which is known to be aversive, is intraorally infused unexpectedly, does MRN serotonergic neurons activate?

2. The authors used the CheRiff-expressing mice and the C128S-expressing mice for the fixed time schedule task. It seems licking during sucrose consumption is more significantly decreased by optogenetic excitation of C128S-expressing neurons than by that of CheRiff-expressing neurons. Would you discuss the difference between the CheRiff-expressing mice and the C128S-expressing mice.

3. In the CheRiff-expressing neurons, 8.6 % is non-serotonin neurons. This means optogenetic stimulation of IPN in the CheRiff-expressing mice may activate not only serotonergic neurons but also vGluT2-expressing neurons. By using the C128S-expressing mice, optogenetic stimulation of IPN would activate almost serotonin neurons. Does IPN serotonin stimulation in the C128S-expressing mice decrease the CPA score?

4. c-Fos expression was observed in the IPN by optogenetic activation of MRN serotonergic neurons in the CheRiff-expressing mice. Does optogenetic activation of MRN serotonergic neurons in the C128S-expressing mice also induce c-fos expression in the IPN?

5. Would you indicate diameter of optic probes used for fiber photometry and optogenetics?

We appreciate the editor and reviewers for providing constructive comments. We have addressed all reviewers' comments and revised the manuscript accordingly (changes are indicated in red font). We believe that this manuscript has been substantially improved.

Responses to Reviewer #1 are as follows:

Comment 1-1.

Reviewer #1 (Remarks to the Author): Comments on NCOMMS-21-44544-T

The brain serotonin system modulates numerous behavioral processes. Serotonin neurons in the dorsal raphe nucleus attract much attention for their roles in reward processing. Although the median raphe nucleus (MRN) is also a major source of serotonin to the forebrain, it has remained elusive whether and how MRN serotonin neurons contribute to reward processing. Previous studies on the MRN often indicate a role of this nucleus and its serotonin neurons in modulating theta rhythm, locomotion, and memory formation (Graeff et al., 1980; Vertes et al., 1994; Kusljic et al., 2003; Kocsis et al., 2006; Fernandez et al., 2017). Here, Kawai et al. examined the potential functions and circuit mechanisms of MRN serotonin neurons in reward processing. Fiber photometry of Ca²⁺ signals indicate that these neurons are inhibited by reward (sucrose) consumption and activated by aversive stimuli (tail pinch). Inhibiting these neurons promotes reward-like behavior, whereas activating the cells produces behavioral aversion. Finally, the authors showed that the projection from the MRN serotonin neurons to the interpeduncular nucleus (IPN) seems particularly important for the behavioral effects. The authors conclude that the serotonergic projections from the MRN, particular the one to the IPN, negatively regulate reward processing.

These results could potentially expand our understanding of the behavioral functions of brain serotonergic system, and may explain how SSRI drugs, while increasing brain serotonin quite rapidly, do not produce strong pleasure in humans.

Ans.

We appreciate the positive evaluation and constructive comments on the manuscript. We have performed additional experiments to address the concerns raised by the reviewer and we feel that our manuscript has been substantially improved through this revision.

Comment 1-2.

I have the major concern on reconciling the well-established functions of the MRN neurons in hippocampal theta, locomotion, and memory formation with the authors' observations and conclusions. Even the IPN, which forms strong connections with the MRN and the nucleus incertus, is

closely involved in modulating hippocampal theta, animal locomotion, and memory processes. The authors did not monitor animal locomotor activity and hippocampal theta during the fiber photometry experiments, so it remains unclear how much changes in Ca²⁺ signals were related to the valence of the stimuli or animal locomotion. The authors did report that activating or inhibiting MRN serotonin neurons did not have a significant effect on overall locomotion (distance travelled). However, the causality could be opposite in direction: locomotor activity changes the neuronal activity of MRN serotonin neuron but not the other way around. Indeed, some of the behavioral effects shown in this study might be interpreted using prior observations that MRN serotonin neurons suppress hippocampal theta. To more convincingly support their conclusions, the authors should simultaneously record neuronal Ca²⁺ signals, hippocampal theta, and animal locomotion in real time. In addition, they should test multiple rewarding and aversive stimuli, each of which has distinct relationship to locomotion. For example, one reward stimulus is associated with locomotion (appetitive chasing), and another with no locomotion (sucrose consumption as shown here). By the same token, one aversive stimulus is associated with locomotion (foot shock), and another with no locomotion (freezing to a pain-predicting neutral cue). This new set of recordings might help unravel how MRN serotonin neurons encode stimulus valence and animal behaving states.

Ans.

We appreciate the reviewer's valuable suggestions. As per the reviewer's suggestion, MRN neurons play a critical role in the hippocampal theta, animal locomotion, and memory processes. During revision, Huang et al. (2022) reported that hippocampal theta oscillations are mainly modulated by non-serotonergic MRN neurons (Huang et al., J Neurosci. 2022). Nevertheless, as the reviewer pointed out, it is possible that locomotor activity changes the activity of MRN serotonin neurons, and Ca²⁺ signal changes in MRN serotonin neurons, which we observed here, may merely reflect locomotor changes rather than valence. To this end, we analyzed the peri-event Ca²⁺ signal in the MRN serotonin neuron before and after termination of locomotion and compared the Ca²⁺ signal in response to sucrose licking in the same animals. We analyzed this pair as licking sucrose solution from a spout requires termination of locomotion. We found a small but significant decrease in Ca²⁺ signal after termination of locomotion (mean stop duration = 2.42 s), but also found that the size of Ca²⁺ signal change in response to sucrose was significantly larger than that to termination of locomotion (**P* = 0.0261; Response Letter Fig. 1). Similarly, we analyzed the peri-event Ca²⁺ signal before and after initiation of locomotion and compared the Ca²⁺ signal in response to tail pinches from the same animals. We analyzed this pair as tail pinching induces locomotion. We found a small but significant increase in Ca²⁺ signal after initiation of locomotion (mean move duration > 3 s) but also found that the size of Ca²⁺ signal change in response to tail pinch was significantly larger than that to initiation of locomotion (***P* = 0.0062; Response Letter Fig. 1).

Response letter Fig. 1. MRN Ca^{2+} signal before and after initiation and termination of locomotion.

Additionally, we measured the Ca^{2+} signal in the MRN in response to quinine solution, a representative aversive stimulus, and vehicle (water). As the tested mice were water-deprived, water was assumed to have a positive valence. We observed a significant decrease in the Ca^{2+} signal in response to water, whereas quinine (5 mM) blunted this decrease (Response letter Fig. 2). Importantly, we found that licking the quinine solution (5 mM) did not induce locomotion during Ca^{2+} signal measurement.

Response letter Fig. 2. MRN Ca^{2+} signal before and after licking of water and quinine solution (5 mM).

Furthermore, we measured the Ca^{2+} signal in MRN serotonin neurons in response to auditory cues associated with foot shocks, using a cue-induced fear conditioning paradigm. Mice were subjected to foot shocks (0.6 mA, 2 s duration, 3 pulses with 60 s intervals). An auditory cue (30 s duration, 2,900 Hz) was applied 28 s before each application of the foot shock and was co-terminated with the foot shock. The next day, the mice were placed in a chamber with different floor textures, different odors (vanilla oil), and different illumination colors. Subsequently, the auditory cue was applied, and the Ca^{2+} signal in the MRN serotonin neuron was measured. We found that the Ca^{2+} signal was significantly increased after the onset of the auditory cue (Response letter Fig. 3).

Response letter Fig. 3. MRN Ca^{2+} signal before and after auditory cue associated with foot shocks.

Collectively, these observations further support our initial findings that MRN serotonin neurons respond to rewarding and aversive stimuli irrespective of locomotor changes. In contrast, we found

that locomotor changes also affected the activity of MRN serotonin neurons, although the effect size was relatively small compared to rewarding and aversive stimuli. This could stem from the number of neurons responding to each stimulus, as fiber photometry measures population activity. Therefore, further analysis using an *in vivo* endoscope, such as nVista, is necessary to determine whether locomotor changes robustly affect the activity of a subpopulation of MRN serotonin neurons.

In relation to hippocampal theta oscillations and memory function, a previous report indicated that the promotion of hippocampal theta oscillations by MRN vGluT2-positive glutamatergic neurons plays a critical role in the formation of fear memory (Szonyi et al., Science. 2019). Therefore, manipulation of MRN serotonin neurons may affect memory formation. Although the conditioned place preference (CPP) and conditioned place aversion (CPA) tests need intact memory function, we found that activation and inhibition induced strong CPA and CPP, respectively, as shown in our original manuscript. This indicates that mice have sufficient memory to associate the conditioned chamber with valence, even in the presence of activation and inhibition of MRN serotonin neurons. We also found that activation of MRN serotonin neurons reduced consummatory licking in the head-fixed setup and hedonic response in facial expression analysis, both of which are much less amenable to confounding memory function. Although these results strongly support our assertion that inhibition and activation of MRN serotonin neurons are rewarding and aversive, respectively, we cannot rule out the possibility that the memory-enhancing effect of manipulation of MRN serotonin neurons overrode CPA and CPP by manipulating the MRN serotonin neurons. Therefore, we discuss this possibility and the importance of determining the role of MRN serotonin neurons in regulating memory formation and hippocampal theta oscillations in the future.

Collectively, we have added the above-mentioned data as Supplementary Figure 1m, n, 2j-p. We describe this point as follows:

(Result)

“As the tested mice should stop locomotion to lick sucrose solution from the spout, licking onset was time-locked to the termination of locomotion. To determine the extent to which changes in GCaMP fluorescence were affected by the termination of locomotion, we analyzed GCaMP fluorescence in the MRN before and after the termination of locomotion and compared the extent of the changes to that of sucrose-induced changes. We found a small but significant GCaMP fluorescence decrease from 0.3 to 1.2 s after the termination of locomotion (Supplementary Fig. 1m), whereas the size of the signal changes (mean $\Delta F/F_0$) after the termination of locomotion was much smaller than that after sucrose licking ($t_6 = 2.935$, $*P = 0.0261$; Supplementary Fig. 1n).”

“As tail pinch increased locomotion, tail pinch was time-locked to the initiation of locomotion. To determine the extent to which changes in GCaMP fluorescence were affected by the initiation of

locomotion, we analyzed GCaMP fluorescence in the MRN before and after the initiation of locomotion and compared the size of the changes to that of tail pinch-induced changes. We found a small but significant GCaMP fluorescence increase after the initiation of locomotion (Supplementary Fig. 2j), whereas the size of the signal changes (mean $\Delta F/F_0$) after the initiation of locomotion was much smaller than that after tail pinch ($t_6 = 4.127$, $**P = 0.0062$; Supplementary Fig. 2k). In addition, we measured GCaMP fluorescence before and after licking with water and quinine solution, another representative aversive stimulus. As the tested mice were water-deprived, water itself was predicted to have a positive value. We observed a significant decrease in GCaMP fluorescence in response to water, whereas quinine (5 mM) blunted this decrease (water: $t_{13} = 2.352$, $*P = 0.0351$; quinine: $t_{13} = 0.2024$, $P = 0.8428$; Supplementary Fig. 2l–o). We then measured GCaMP fluorescence in MRN serotonin neurons in response to auditory cues associated with foot shocks, using a cue-induced fear conditioning paradigm³⁸. One day after training, GCaMP fluorescence in the MRN was measured before and after the onset of the auditory cues associated with foot shocks. We found that GCaMP fluorescence increased after the onset of the auditory cue ($P < 0.05$, $n = 5$ mice; Supplementary Fig. 2p).

(Discussion)

“Previous reports have shown that MRN neurons play a critical role in the regulation of hippocampal theta oscillations, locomotion, and memory function^{57,71,72,73,74} as well as in the processing of appetitive and aversive stimuli. Although the extent of suppression was much smaller than that of sucrose, we found that immobility induced a significant decrease in GCaMP fluorescence in the MRN. Similarly, the initiation of locomotion induced a significant increase in GCaMP fluorescence in the MRN, although the size of the increase was substantially smaller than that of the tail pinch (Supplementary Fig. 1m, n, 2j, k). Additionally, quinine, another aversive stimulus^{16,17}, blunted water-induced decreases in GCaMP fluorescence in MRN. Furthermore, auditory cues associated with foot shocks, which induce immobility in a new context³⁸, increased GCaMP fluorescence in the MRN. Collectively, these results indicate that MRN serotonergic neurons respond to rewarding and aversive stimuli irrespective of locomotor changes. At the same time, our data indicate that locomotor activity affects the activity of MRN serotonergic neurons, although the size of the changes was relatively small. This could stem from the number of neurons responding to each stimulus, as fiber photometry measures the fluorescence of a whole nucleus as a proxy for population activity. Therefore, further analysis using *in vivo* endoscopy is required to determine whether locomotor changes robustly affect the activity of a subpopulation of MRN serotonergic neurons. Although a recent report has indicated that hippocampal theta oscillations are mainly modulated by non-serotonergic MRN neurons⁷⁵, it is possible that the manipulation of MRN serotonergic neurons affects memory formation. However, we found that activation and inhibition induced strong CPA and CPP, both of which required intact

memory function. These results indicate that mice have sufficient memory to associate the conditioned chamber with valence, even in the presence of manipulation of MRN serotonergic neurons. We also found that activation of MRN serotonin neurons reduced consummatory licking in the head-fixed setup and hedonic response in facial expression analysis, both of which are much less amenable to confounding memory function. Although these results strongly support our notion that inhibition and activation of MRN serotonergic neurons are rewarding and aversive, respectively, we cannot rule out the possibility that the memory-enhancing effect of manipulation of MRN serotonergic neurons overrode CPA and CPP by manipulating MRN serotonergic neurons. Further analysis of the role of MRN serotonin neurons in regulating memory formation and hippocampal theta oscillation will be crucial for a better and comprehensive understanding of the role of MRN serotonergic neurons.”

Comment 1-3.

I have other more specific concerns:

1. Figure 4 shows strong terminal labeling in the VTA and the PVT, as well as some clear labeling in the LHb. All of these areas are involved in reward processing. The authors instead focused on the ventral hippocampus, IPN, and MHb to examine the potential downstream targets of MRN serotonin neurons. Please explain the rationale behind the choices and how the VTA and PVT might not be important despite the dense innervations.

Ans.

We apologize for the confusing figure. In the VTA, we observed strong SERT expression, but weak eGFP fluorescence derived

Fig. 4b of the original manuscript.

Response letter Fig. 4. Ca^{2+} signal changes of MRN serotonin neuron terminals in the LHb (Top) and PVT (Bottom) in response to sucrose licking.

from MRN serotonin neurons (Fig. 4b). Taken together with previous reports analyzing projection of DRN serotonin neuron (Muzerelle et al., Brain Struct Func. 2016; Nagai et al., Int J Mol Sci. 2020), we considered these SERT-positive fibers as axons of the DRN serotonin neurons. As the reviewer pointed out, we observed clear eGFP expression in the PVT and the LHb. Therefore, we measured Ca^{2+} signals in MRN serotonin neuronal fibers in the PVT and LHb in response to sucrose licking. We found no significant difference in the Ca^{2+} signal in response to sucrose in the LHb and PVT between Venus- and GCaMP6-expressing animals (Response letter Fig. 4). These data highlight the importance of the IPN and vHP in the processing of reward information. We have added these data to Supplementary Fig. 6s–v and described this result in the Results section as follows:

“Moreover, we measured axon-GCaMP fluorescence in the lateral habenula (LHb) and the paraventricular nucleus of the thalamus (PVT), where clear projections of MRN serotonergic neurons were observed (Fig. 4b, d). However, we did not find significant fluorescence changes in either nucleus in response to sucrose consumption compared with the Venus control (Supplementary Fig. 6s–v).”

Comment 1-4

2. *In addition, the IPN is adjacent to the MRN, and the serotonergic fibers from the MRN likely pass through the IPN en route to forebrain centers through the fasciculus retroflexus. This may explain how stimulating or inhibiting the IPN produces the strongest effect. This spatial closeness may even confound data interpretation for the fiber photometry and retrograde labeling experiments.*

3. *The authors attempted to confirm the validity of optogenetic stimulation by showing more c-Fos+ cells in the IPN of CheRiff-stimulated mice (Extended Data Fig. 5). This approach is problematic, since the authors showed that stimulating MRN serotonin neurons did not increase c-Fos expression in the vHP but manipulating the terminals of these neurons in the vHP changed the reward and aversion processing. Therefore, c-Fos is not a very good marker of neuronal activation here.*

Ans.

We thank the reviewer for this valuable suggestion. In relation to the latter point, we have performed electrophysiological analyses (shown in response to comment 1-5). In addition, we analyzed c-Fos expression in pyramidal and non-pyramidal neurons in the vHP separately. We found that optogenetic stimulation of MRN serotonin neurons (soma stimulation) significantly increased the number of c-Fos-positive non-pyramidal neurons, indicating that c-Fos expression in non-pyramidal neurons could be used as a marker for activation of serotonergic terminals in the vHP (Response letter Fig. 5). Under these conditions, we performed optogenetic stimulation of $5-HT^{MRN \rightarrow IPN}$ and counted c-Fos-

expressing cells in the IPN and vHP to determine the extent to which 5-HT^{MRN→IPN} stimulation affects serotonergic terminals in the vHP. We found that stimulation significantly increased the number of c-Fos-expressing cells in the IPN but did not affect the number of c-Fos-positive non-pyramidal neurons in the vHP (Response letter Fig. 6). Furthermore, there was no significant increase in c-fos-expressing serotonin neurons in the MRN after optogenetic stimulation with 5-HT^{MRN→IPN}, despite the spatial closeness between the MRN and IPN (Response letter Fig. 6). These results indicate successful stimulation of 5-HT^{MRN→IPN} with minimal intervention of serotonergic projections from the MRN to the vHP. Nevertheless, we cannot rule out the possibility that stimulation of 5-HT^{MRN→IPN} may affect other serotonergic projections from the MRN, clearly indicating the importance of brain-wide activity mapping for the unbiased identification of downstream targets of MRN serotonergic neurons in the future. We describe these results in Supplementary Fig. 5 as follows:

(Result)

“However, when we analyzed c-Fos-positive pyramidal and non-pyramidal neurons in the vHP separately, we found significantly more c-Fos-positive non-pyramidal neurons in CheRiff

mice than in Venus mice, whereas there was no significant difference between groups in c-Fos-positive pyramidal neurons (vHP (non-pyramidal): $n = 6$ (Venus) and $n = 9$ (CheRiff), $t_{13} = 3.806$, $**P = 0.0022$, Supplementary Fig. 5g; vHP (pyramidal): $n = 6$ (Venus) and $n = 9$ (CheRiff), $t_{13} = 0.4284$, $P = 0.6754$, Supplementary Fig. 5h). These results indicate that optogenetic activation of MRN serotonergic neurons increased neuronal activity in the IPN and vHP.”

Response letter Fig. 5. Optogenetic stimulation of MRN serotonergic neurons increased the number of c-Fos-expressing non-pyramidal neurons in the vHP

Response letter Fig. 6. Optogenetic stimulation of 5-HT^{MRN→IPN} preferentially increased the number of c-Fos-expressing cells in the IPN

“Considering spatial closeness between IPN and MRN and the presence of serotonergic fibers passing through the IPN, optogenetic stimulation in the IPN may affect serotonergic projections from the MRN to other brain regions including vHP, which leads to an apparently strong effect induced by IPN stimulation. Therefore, we performed optogenetic stimulation of 5-HT^{MRN→IPN} and counted c-Fos-expressing cells in the IPN and vHP to determine the extent to which 5-HT^{MRN→IPN} stimulation affects serotonergic terminals in the vHP. Stimulation significantly increased the number of c-Fos-expressing cells in the IPN, but did not affect the number of c-Fos-positive non-pyramidal neurons in the vHP (Supplementary Fig. 5l, m, o, p, q). Furthermore, there was no significant increase in c-Fos-expressing MRN serotonergic neurons after optogenetic stimulation with 5-HT^{MRN→IPN} (Supplementary Fig. 5n). These results indicate successful stimulation of 5-HT^{MRN→IPN} with minimal intervention to other serotonergic projections, including 5-HT^{MRN→vHP}.”

(Removed from Result)

“Moreover, optogenetic activation of MRN serotonergic neurons increased the total number of c-Fos-positive cells in the IPN but not in the vHP or MHb (Extended Data Fig. 5).”

(Discussion)

“Optogenetic activation of MRN serotonergic neurons increased the number of c-Fos-positive cells in the IPN and non-pyramidal neurons in the vHP. Additionally, we found that stimulation of 5-HT^{MRN→IPN} significantly increased the number of c-Fos-expressing cells in the IPN, but did not affect the number of c-Fos-positive non-pyramidal neurons in the vHP and c-Fos-positive cells in the MRN. These results indicate successful stimulation of 5-HT^{MRN→IPN} with minimal intervention in 5-HT^{MRN→vHP}. Nevertheless, we cannot rule out the possibility that stimulation of 5-HT^{MRN→IPN} may affect other serotonergic projections from the MRN, indicating the importance of brain-wide activity mapping for unbiased identification of downstream targets of MRN serotonergic neurons in the future.”

As the reviewer mentioned, it is possible that the spatial closeness between the IPN and MRN and the presence of serotonergic fibers passing through the IPN confound data interpretation of photometry measurement and retrograde labeling. For photometry measurements, the intensity of the excitation light was weaker by two orders of magnitude than that of optogenetic manipulation (several tens of microwatts vs. several milliwatts). Hence, it is less likely that axonGCaMP fluorescence in the MRN affected axonGCaMP fluorescence in the IPN. In contrast, we cannot rule out the possibility that the axonGCaMP fluorescence of serotonergic fibers passing through the IPN affects that in the IPN. However, a previous report has shown that the fluorescence changes of axonGCaMP6 in presynaptic

boutons are two times larger than those in axon shafts, whereas there was no difference in the baseline fluorescence of axonGCaMP6 in presynaptic boutons and axon shafts (Supplementary Fig. 7 of Broussard et al., Nat Neurosci. 2018). Hence, the larger the contribution of baseline fluorescence derived from the fiber passing through the IPN, the less the contribution of axonGCaMP fluorescence of the presynaptic bouton in the IPN (i.e., 5-HT^{MRN→IPN}) to changes in fluorescence. If changes in fluorescence in the IPN may largely originate from the axon shaft passing through the IPN, there should be other brain regions where much larger fluorescent changes are observed than in the IPN. However, we found that changes in fluorescence (relative to baseline fluorescence (F_0)) in the IPN were much larger than those in other brain regions, including vHP, MHb, LHb, and PVT (as shown above), where dense MRN serotonergic projections were observed. Collectively, these results indicate a relatively large contribution of 5-HT^{MRN→IPN} to the fluorescence changes in the IPN. Nevertheless, we cannot completely rule out the possibility that there may be other MRN serotonergic projections responding to appetitive and aversive stimuli. Thus, we acknowledged this limitation in the revised version of our manuscript.

For retrograde labeling, previous reports have indicated that Retrobeads enable retrograde labeling with minimal entry into undamaged fibers of passage and minimal diffusion at the injection site (Saleeba et al., Front Neurosci. 2019), indicating that retrograde labeling from the IPN to the MRN serotonergic neurons, which we observed here, was mainly due to 5-HT^{MRN→IPN}. We describe the aforementioned points as follows:

(Abstract)

(from) “We further identified MRN serotonergic neurons projecting to the interpeduncular nucleus (5-HT^{MRN→IPN}) as a key mediator of reward and aversive stimuli.”

(to) “We further identified MRN serotonergic neurons, including those projecting to the interpeduncular nucleus (5-HT^{MRN→IPN}), as a key mediator of reward and aversive stimuli.”

(from) “Our findings revealed an essential function of 5-HT^{MRN→IPN} in the processing of reward and aversive stimuli, thus providing insights into the partial efficacy of SSRIs for anhedonia.”

(to) “Our findings revealed an essential function of MRN serotonergic neurons, including 5-HT^{MRN→IPN}, in the processing of reward and aversive stimuli, thus providing insights into the partial efficacy of SSRIs in anhedonia.”

(Discussion)

“Although our results strongly suggest a key role of 5-HT^{MRN→IPN}, there is a limitation of our study, mainly due to the presence of serotonergic fibers passing through the IPN and the spatial closeness between the IPN and MRN. Considering the spatial closeness of the IPN and MRN, it is possible that

GCaMP fluorescence in the MRN may affect photometry measurements in the IPN. However, histochemical analysis revealed that optogenetic manipulation in the IPN with much more intense excitation light than photometry measurements did not affect the activity in the MRN (Supplementary Fig. 5n). Hence, it is less likely that axon-GCaMP fluorescence in the MRN affected axon-GCaMP fluorescence in the IPN. In contrast, we cannot rule out the possibility that the axon-GCaMP fluorescence of serotonergic fibers passing through the IPN affects that in the IPN. However, a previous report has shown that fluorescent changes in axon-GCaMP6 in presynaptic boutons are twice as large as those in the axon shaft, whereas there was no difference in the baseline fluorescence of axon-GCaMP6 in presynaptic boutons and axon shaft⁵². Hence, the larger the contribution of baseline fluorescence derived from the fiber passing through the IPN, the less the contribution of axon-GCaMP fluorescence of presynaptic boutons in the IPN (i.e., 5-HT^{MRN→IPN}) to fluorescence changes. If fluorescence changes in the IPN may largely originate from the axon shaft passing through the IPN, there should be other brain regions where much larger fluorescent changes are observed than in the IPN. However, we found that fluorescence changes (relative to baseline fluorescence (F_0)) in the IPN were much larger than those in other brain regions, including vHP, MHb, LHb, and PVT (Fig. 5, 6, Supplementary Fig. 6s–v), where dense MRN serotonergic projections were observed. Collectively, these results indicate a relatively large contribution of 5-HT^{MRN→IPN} to the fluorescence changes in the IPN. Nevertheless, we cannot completely rule out the possibility that other MRN serotonergic projections respond to appetitive and aversive stimuli. For retrograde labeling, previous reports have indicated that Retrobeads enable retrograde labeling with minimal entry into undamaged fibers of passage and minimal diffusion at the injection site⁸⁶, indicating that retrograde labeling from the IPN to the MRN serotonergic neurons, which we observed here, was mainly due to 5-HT^{MRN→IPN}.”

(from) “In conclusion, we identified MRN serotonergic neurons, especially those projecting to the IPN, as important mediators in the processing of rewards and aversive stimuli.”

(to) “In conclusion, we identified MRN serotonergic neurons, including those projecting to the IPN, as important mediators in the processing of rewards and aversive stimuli.”

Comment 1-5

Importantly, the authors did not provide any good evidences that their choices of parameters for optogenetic interventions effectively changed neuronal activity in vivo or ex vivo (brain slices). I am particularly concerned about the choice of eArchT, which in our hands may not be very effective in suppressing neuronal firing in some brain areas but at the same time may induce post-inhibition rebound.

Ans.

We appreciate the reviewer for this important suggestion. According to the comment, we performed *ex vivo* electrophysiological measurement. We found that blue LED light stimulation (20 Hz frequency, 10 ms duration, 40 pulses, 5.92 mW) evoked action potentials with $80.4\% \pm 5.5\%$ fidelity (32.2 ± 2.2 action potentials / 40 pulses, $n = 5$ cells from 2 mice) in CheRiff-expressing animals (Response letter Fig. 7a, b). We also found that green LED light stimulation (3 s duration, 3.5 s after the start of depolarizing current injection, 1.58 mW) induced hyperpolarization, which suppressed firing activities induced by a depolarizing current injection (0-30 pA, 10 s duration) (frequency of action potentials (mean \pm s.e.m.); before, 2.0 ± 0.33 Hz, light stimulation, 0.038 ± 0.0028 Hz, after, 1.5 ± 0.33 Hz; one-way repeated measures ANOVA with Bonferroni's *post hoc* test, $F_{1,69/8,47} = 22.52$, $***P = 0.0005$; *post hoc* test: before vs light, $**P = 0.0062$, before vs after, $P = 0.2209$, light vs after, $*P = 0.0221$, 30 sweeps in 6 cells from 3 mice) in eArchT-expressing animals (Response letter Fig. 7c, d). Importantly, we did not observe significant difference in activity before and after optogenetic inhibition at least under this condition, indicating that rebound activation was not induced by optogenetic inhibition. These results indicate successful optogenetic manipulation of MRN serotonergic neurons. We described these results in Supplementary Fig. 1o, p, 2q, r and as follows:

Response letter Fig. 7. Electrophysiological analysis of optogenetic manipulation. (a, b) CheRiff (c, d) eArchT. Scale bar = 30 μ m.

(Result)

“*Ex vivo* electrophysiology experiment revealed that activity of eArchT-expressing cells was significantly attenuated by green light illumination and was reversed to the basal level after light illumination (before: 2.0 ± 0.33 Hz, light: 0.038 ± 0.0028 Hz, after: 1.5 ± 0.33 Hz, 30 sweeps in 6 cells from 3 mice; $***P = 0.0005$ by one-way repeated measures ANOVA, $**P = 0.0062$ (before vs. light), $P = 0.2209$ (before vs. after), $*P = 0.0221$ (light vs. after) by paired *t*-test, Supplementary Fig. 1o, p).”

“*Ex vivo* electrophysiology experiment revealed that blue light illumination induced firing of CheRiff-expressing cells, which was time-locked to the illumination ($80.4\% \pm 5.5\%$ fidelity (32.2 ± 2.2 action potentials/40 pulses, $n = 5$ cells from 2 mice, Supplementary Fig. 2q, r).”

Comment 1-6

4. The authors showed that intra-IPN administration of MDL-100907 reversed the reduction in the CPA score induced by MRN activation (Figure 7). The authors need to validate 5-HT_{2A} expression by immunostaining. Due to the close distance between the IPN and the VTA, MDL-100907 might diffuse to the VTA and thus blocked the 5-HT_{2A} receptors there. Moreover, there are multiple serotonin receptors that are richly expressed in the IPN, either in the axonal terminals of MHB neurons (e.g. 5-HT₄) or in IPN neurons. The authors need to take great caution when making the statement that "5-HT_{2A} receptors in the IPN mediate the aversive properties associated with MRN serotonergic neuron activity". Nowadays pharmacological evidences are often backed up by in vivo genetic manipulations, such as Crispr-Cas9-mediated knockout of 5-HT_{2A} in the IPN here, to reach a more clear-cut conclusion.

Ans.

We appreciate the reviewer's valuable comments. We apologize for the confusing figure showing the MRN serotonergic projections in the IPN. As described in the responses to comment 1-3 above, we considered the SERT-positive fibers in the VTA without eGFP derived from MRN serotonergic neurons as axons of DRN serotonergic neurons. In accordance with this comment, we checked the expression of 5-HT_{2A} in the IPN using immunostaining and RT-PCR to validate 5-HT_{2A} expression. We found immunoreactivity for two different 5-HT_{2A} receptor antibodies (Alomone ASR-033 and ImmunoStar 24288) in the IPN, whereas a relatively weak 5-HT_{2A} immunoreactivity was observed in the VTA. Moreover, we found cell soma-like immunoreactivity for 5-HT_{2A} receptors in the IPN that did not colocalize with the serotonin transporter, indicating the postsynaptic expression of 5-HT_{2A} receptors (Response letter Fig. 8). The expression of 5-HT_{2A} mRNA was further confirmed by RT-PCR analysis of cDNA prepared from mouse IPN (Response letter Fig. 8).

Alomone (ASR-033)

RT-PCR of IPN cDNA

Immunostar (#24288)

Gapdh marker
Htr2a (5-HT_{2A}) rep1 rep2
Htr2b (5-HT_{2B}) rep1 rep2

Predicted Size 75 bp 97 bp 84 bp

Response letter Fig. 8.
 Immunohistochemical and RT-PCR
 analysis of 5-HT_{2A} receptors in the
 IPN

In addition, we examined the effects of intra-IPN administration of TCB-2, a 5-HT_{2A} agonist, on CPA scores without optogenetic manipulation. We found that intra-IPN administration of TCB-2 (0.05 μg/0.5 μL) significantly decreased the time spent in the chamber associated with TCB-2 (Response letter Fig. 9). Although these results further indicate the possible involvement of IPN 5-HT_{2A} receptors, it is possible that other serotonin receptors expressed in the IPN play a critical role, as the reviewer suggested. Therefore, we have modified the statements regarding 5-HT_{2A} receptors in the IPN throughout the manuscript and the importance of further experiments using gene knockdown and

Response letter Fig. 9. Effect with
 intra-IPN injection of TCB-2, a 5-
 HT_{2A} agonist, on CPA score.

Cas9-mediated knockout in the IPN as follows:

(Abstract)

(from) “Moreover, the aversive properties of MRN serotonergic neural activity required activation of 5-HT_{2A} receptors in the interpeduncular nucleus.”

(to) “Moreover, 5-HT receptors, including the 5-HT_{2A} receptors in the interpeduncular nucleus, are involved in the aversive properties of MRN serotonergic neural activity.”

(Result)

(from) “Activation of 5-HT_{2A} receptors in the IPN is necessary for the aversive effect of MRN serotonergic neuron activation”

(to) “Activation of 5-HT receptors including 5-HT_{2A} receptors in the IPN is necessary for the aversive effect of MRN serotonergic neuron activation”

“Immunohistochemical analysis using two different 5-HT_{2A} receptor antibodies revealed immunoreactivity with cell soma-like morphology in the IPN (Supplementary Fig. 10g, h). We also found 5-HT_{2A} mRNA expression in the IPN using RT-PCR (Supplementary Fig. 10i). We examined the effects of intra-IPN administration of TCB-2, a 5-HT_{2A} agonist, on CPA scores without optogenetic manipulation. We found that intra-IPN administration of TCB-2 (0.05 µg/0.5 µL) significantly decreased the CPA score and the time spent in the chamber associated with TCB-2 (Fig. 7i, j and Supplementary Fig. 10e, f).”

(from) “Taken together, these results demonstrated the key role of 5-HT_{2A} receptors in the IPN in the processing of aversive stimuli.”

(to) “Taken together, these results indicate that 5-HT receptors, including 5-HT_{2A} receptors in the IPN, are involved in the processing of aversive stimuli.”

(Discussion)

(from) “Moreover, we showed the necessity of 5-HT_{2A} receptor activation in the IPN for the acquisition of conditioned place aversion.”

(to) “Moreover, we showed the involvement of 5-HT receptors, including 5-HT_{2A} receptors in the IPN, for the acquisition of conditioned place aversion.”

“In addition to 5-HT_{2A} receptors, previous reports have shown strong expression of several 5-HT receptors including 5-HT_{1A}⁶¹, 5-HT_{1B}⁶², and 5-HT₄⁶³. Considering the limitation of pharmacological intervention with regard to receptor selectivity, further analysis using gene knockdown and Cas9-

mediated *in vivo* gene knockout⁶⁴ is necessary for determining the 5-HT receptors critical for processing aversive stimuli.”

(from) “Moreover, our data suggested that 5-HT_{2A} receptors in the IPN mediate the aversive properties associated with MRN serotonergic neuron activity.”

(to) “Moreover, our data suggested that 5-HT receptors, including 5-HT_{2A} receptors in the IPN, are involved in the aversive properties associated with MRN serotonergic neuron activity.”

Comment 1-7

Minor concerns:

1. The efficiencies of viral expression are critical for the reliabilities of experiments. We recommend the authors to verify the expression efficiencies of every virus batch and check the rates of transgenes expression in serotonin neurons.

Ans.

We thank the reviewer for this valuable suggestion. We used the same batch of viruses throughout this study. Immunohistochemical analysis was used to check whether the GCaMP virus has similar specificity to the Venus virus, with which we checked the specificity to serotonergic neurons. We found that GCaMP and axon-GCaMP viruses have similar specificity as that of the Venus virus ((GCaMP6s) specificity 95.9% ± 0.9%, coverage 92.0% ± 2.3%; (axon-GCaMP6s) specificity 94.4% ± 1.0%, coverage 90.7% ± 2.7%). Due to the membrane localization of CheRiff and eArchT, it was not possible to reliably measure the specificity and coverage of these AAVs. We have described this result in the Materials and Methods.

“We confirmed that efficiency of AAV bearing GCaMP6s and axon-GCaMP6s were similar to that of the AAV bearing Venus ((GCaMP6s) specificity 95.9% ± 0.9%, coverage 92.0% ± 2.3%; (axon-GCaMP6s) specificity 94.4% ± 1.0%, coverage 90.7% ± 2.7%). Owing to the membrane localization of CheRiff and eArchT, it was not possible to reliably measure the specificity and coverage of these AAVs.”

Comment 1-8

2. In Fig 1h, the GCaMP signals increase before the sucrose licking. Does this mean that reward anticipation increases the activities of MRN serotonin neurons?

Ans.

We appreciate the reviewer's suggestion. According to the comment, we reanalyzed the data under the same conditions as the DRN data, where F_0 was defined as the mean fluorescence from -2 to -0.5 s. We found a significant increase in Ca^{2+} signal from -1 to -0.5 s as well a decrease after sucrose licking, as the reviewer mentioned (Response letter Fig. 10). Because the tested mice had experienced a number of sucrose licks before measurement to acclimate to the setup, further measurement using appetitive Pavlovian conditioning, as

Response letter Fig. 10. GCaMP fluorescence in MRN serotonin neuron with baseline setting used in DRN serotonin neuron.

previously reported in the DRN (Zhong et al., J Neurosci. 2017), is required to determine whether this Ca^{2+} signal increase in MRN serotonin neurons is associated with reward anticipation. We have added this result to the Supplementary Fig. 1i-l, and describe the results and discussion as follows:

(Results)

“When we analyzed the GCaMP fluorescence changes in the same baseline setting as the previous report on DRN serotonergic neurons¹⁶, we found a significant increase in GCaMP fluorescence from -1 to -0.5 s as well as a decrease after sucrose licking (Supplementary Fig. 1i-l).”

(Discussion).

“Through analysis of MRN GCaMP fluorescence in the same baseline setting as that of a previous report on DRN serotonergic neurons¹⁶, we found increased GCaMP fluorescence before sucrose licking. As the tested mice had experienced a number of sucrose licks before measurement to acclimate to the setup, further measurement using appetitive Pavlovian conditioning, as previously reported in the DRN¹⁷, is needed to determine whether increased activity in MRN serotonin neurons is associated with reward anticipation.”

Comment 1-9

3. *The title of Extended Data Fig.5 “c-Fos expression in the MRN was induced by light stimulation in IPN, vHP, and MHb”, is consistent with the description in the main text.*

Ans.

We apologize for the incorrect title in Extended Data Fig. 5. Because we added the data as described in response to Comment 1-4, we corrected the title to “c-Fos expression in the IPN, vHP, MHb, and MRN after light stimulation.”

Responses to Reviewer #2 are as follows:

Comment 2-1

Reviewer #2 (Remarks to the Author):

In this study, Kawai and colleagues show the role of MRN in processing reward and aversive information. They find that

1. MRN serotonergic neurons increase and decrease their activity by an aversive stimulus tail pinch and an appetitive stimulus sucrose solution, respectively.

2. Optogenetic excitation and inhibition of MRN serotonergic neurons induce aversive effect measured by the conditioned place aversion (CPA) test and reward effect measured by the conditioned place preference (CPP) test, respectively.

3. MRN serotonergic neurons projecting to the IPN and the vHP increase and decrease their activity by the tail pinch and sucrose solution consumption, respectively.

4. Optogenetic excitation and inhibition of MRN serotonergic neurons projecting to the IPN and the vHP induce aversive effect measured by CPA test and reward effect measured by the CPP test, respectively.

5. 5-HT_{2A} receptors in the IPN are necessary in the processing of aversive stimuli.

These results show the precise mechanism how MRN serotonergic neurons work in processing reward and aversive stimuli. The finding that MRN serotonergic neurons are activated by aversive stimuli is contrast to DRN serotonergic neurons which are mainly activated by reward stimuli. This study provides very important data. I have a few issues to be clarified for aversive processing of MRN serotonergic neurons.

Ans.

We appreciate the reviewer for a positive evaluation of our manuscript and constructive comments.

Comment 2-2

1. I am very interested in whether MRN serotonergic neurons respond to other aversive stimuli other than the tail pinch. The tail pinch induces pain. Does the CPA score decrease because optogenetic activation of MRN serotonergic neurons causes pain? For example, when quinine solution, which is known to be aversive, is intraorally infused unexpectedly, does MRN serotonergic neurons activate?

Ans.

We thank the reviewer for this valuable suggestion. We measured Ca^{2+} signals in MRN serotonin neurons in response to spontaneous drinking of the quinine solution. As the tested mice were water-deprived, we found that water significantly decreased GCaMP fluorescence in the MRN, whereas quinine (5 mM) blunted this decrease (Response letter Fig. 2). Additionally, we measured the Ca^{2+} signal in MRN serotonin neurons in response to auditory cues associated with foot shock, using a cue-induced fear conditioning paradigm. Mice were subjected to foot shocks (0.6 mA, 2 s duration, 3 pulses with 60 s intervals). An auditory cue (30 s duration, 2,900 Hz) was applied 28 s before each application of the foot shock and was co-terminated with the foot shock. The next day, the mice were placed in a chamber with different floor textures, different odors (vanilla oil), and different illumination colors. Subsequently, an auditory cue was applied, and the Ca^{2+} signal in the MRN serotonin neuron was measured. We found that the Ca^{2+} signal increased after the application of the auditory cue (Response letter Fig. 3). Collectively, these results indicate that the activation of MRN serotonin neurons is induced by aversive stimuli, with and without pain. We described this result in Supplementary Fig. 2l–p as follows:

Response letter Fig. 2. MRN Ca^{2+} signal before and after licking of water and quinine solution (5 mM).

(Result)

“In addition, we measured GCaMP fluorescence before and after licking with water and quinine solution, another representative aversive stimulus. As the tested mice were water-deprived, water itself was predicted to have a positive value. We observed a significant decrease in GCaMP fluorescence in response to water, whereas quinine (5 mM) blunted this decrease (water: $t_{13} = 2.352$, $*P = 0.0351$; quinine: $t_{13} = 0.2024$, $P = 0.8428$; Supplementary Fig. 21–o). We then measured GCaMP fluorescence in MRN serotonin neurons in response to auditory cues associated with foot shocks, using a cue-induced fear conditioning paradigm³⁸. One day after training, GCaMP fluorescence in the MRN was measured before and after the onset of the auditory cues associated with foot shocks. We found that GCaMP fluorescence increased after the onset of the auditory cue (Supplementary Fig. 2p).”

Response letter Fig. 3. MRN Ca^{2+} signal before and after auditory cue associated with foot shocks.

Comment 2-3

2. The authors used the CheRiff-expressing mice and the C128S-expressing mice for the fixed time schedule task. It seems licking during sucrose consumption is more significantly decreased by optogenetic excitation of C128S-expressing neurons than by that of CheRiff-expressing neurons. Would you discuss the difference between the CheRiff-expressing mice and the C128S-expressing mice.

Ans.

We agree with your opinion that C128S-expressing transgenic mice show stronger effects. We speculate that this is because transgenic mice express C128S in virtually all serotonin neurons (Ohmura et al. 2014, Int J Neuropsychopharmacol) whereas CheRiff-expressing mice express CheRiff only in AAV-infected neurons. The same AAV promoter achieved a high specificity and coverage (> 90%; Fig 1d). However, it should be noted that we did not inspect the entire MRN but counted the number of serotonin neurons expressing Venus around the injection site. Although it was an approximate calculation, we speculate that about half of the serotonin neurons in the MRN expressed CheRiff as we injected AAV into the rostral part of the MRN (Supplementary Figure 2i), and AAV spread approximately 0.7–1 mm around the site of injection. We discuss this point as follows.

“The reduction of licking behavior looks more prominent in the bigenic mice than in CheRiff mice. As we injected AAV into the rostral part of the MRN in CheRiff mice, it is likely that serotonergic neurons in the caudal part of the MRN were more faithfully manipulated in bigenic mice.”

Comment 2-4

3. In the CheRiff-expressing neurons, 8.6 % is non-serotonin neurons. This means optogenetic stimulation of IPN in the CheRiff-expressing mice may activate not only serotonergic neurons but also vGluT2-expressing neurons. By using the C128S-expressing mice, optogenetic stimulation of IPN would activate almost serotonin neurons. Dose IPN serotonin stimulation in the C128S-expressing mice decrease the CPA score?

Ans.

We appreciate the reviewer’s suggestion. We examined the effect of optogenetic stimulation of the IPN on CPA scores in C128S-expressing mice. We found that the time spent in the chamber associated with optogenetic stimulation (500 ms duration, 0.1 Hz, 20 min) in C128S-expressing mice was

significantly shorter than that in control mice (Response letter Fig. 11). These results further support the critical role of IPN-projecting 5-HT neurons in the MRN. The results are shown in Supplementary Fig. 8n, o, as follows:

“Additionally, optogenetic stimulation in the IPN of the bigenic mice significantly decreased the CPA score and the time spent in the chamber associated with optogenetic stimulation ($n = 7$ (control) and $n = 7$ (bigenic mice), (CPA score) $t_{12} = 2.686$, $*P = 0.020$, (spent time-ChR2 bigenic) $t_6 = 4.212$, $**P = 0.0056$, Supplementary Fig. 8n, o).”

Response letter Fig. 11. Optogenetic stimulation of the IPN in C128S-expressing mice induced CPA.

Comment 2-5

4. *c-Fos* expression was observed in the IPN by optogenetic activation of MRN serotonergic neurons in the *CheRiff*-expressing mice. Does optogenetic activation of MRN serotonergic neurons in the C128S-expressing mice also induce *c-fos* expression in the IPN?

Ans.

We appreciate your suggestion. After optogenetic stimulation of the MRN in C128S-expressing mice, we quantified the number of *c-Fos*-expressing cells in the IPN. Histochemical analysis revealed that the number of *c-Fos*-positive cells in C128S-expressing mice was significantly larger than that in the control (Response letter Fig. 12). This result indicates that the optogenetic stimulation of MRN serotonergic neurons increases IPN activity. We describe this result in Supplementary Fig. 8m as follows:

C128S-expressing mice
optogenetic stimulation in the MRN

Response letter Fig. 12. Optogenetic stimulation of the MRN in C128S-expressing mice increased the number of *c-Fos*-expressing cells in the IPN.

“In Tph2-tTA::tetO-ChR2(C128S)-eYFP bigenic mice where virtually all ChR2-expressing neurons were serotonergic, we found significantly more c-Fos-positive cells in IPN of the bigenic mice than control mice after optogenetic stimulation in the MRN ($n = 5$ (control) and $n = 4$ (bigenic mice), $t_7 = 4.21$, $**P = 0.004$, Supplementary Fig. 8m).”

Comment 2-6

5. Would you indicate diameter of optic probes used for fiber photometry and optogenetics?

Ans.

The diameter of the optic probes used for fiber photometry and optogenetics were 250 μm /240 μm (Clad/Core). We have added this information to Materials and Methods.

Reviewers' Comments:

Reviewer #1:

Remarks to the Author:

I appreciate the great efforts that the authors have made in response to my questions and concerns. The manuscript has been substantially improved. I am for most part satisfied with their revisions. Still, I would like to invite the authors to address the two remaining concerns.

1. The authors concluded that the projection to the interpeduncular nucleus (IPN) underlies the aversive processing role of median raphe (MRN) 5-HT neurons. To support this conclusion, they used the inefficacy of the MRN projection to the medial habenula (MHb) as the negative example of non-IPN projections. However, previous tracing studies and Fig. 4 in the current study clearly show that MRN 5-HT neurons provide much denser output to the paraventricular thalamus (PVT) and the lateral habenula (LHb) than that to the MHb. In addition, numerous studies have well established that the PVT and the LHb participate in aversive processing. The conclusion would be much stronger if the authors can experimentally demonstrate that the serotonergic projections from the MRN to the PVT and the LHb are indeed not involved in aversive processing. At the very minimum, the authors should spell out the limitations of their current dataset and discuss the potential involvements of the MRN projections to other brain areas, including the PVT and LHb.

2. The authors now showed the change of Ca²⁺ signals in relation to the locomotor onset and offset. They conclude that MRN 5-HT neurons exhibit a weak but significant response to locomotor activity. However, they did not align the Ca²⁺ signals to change of locomotor speed, so it is unclear whether weak signals were due to small locomotor activity, and whether the signals will be substantially stronger if mice walked more rapidly. I suggest the authors to further discuss this possibility.

Reviewer #2:

Remarks to the Author:

The authors have satisfactorily addressed my comments. This is a very interesting and important study.

We appreciate the editor and reviewers for providing constructive comments. We have revised the manuscript according to the reviewers' comments (changes are indicated in red font for reviewers' comments and in blue font for editor's comments). We believe that this manuscript has been substantially improved.

Responses to Reviewer #1 are as follows:

Comment 1-1.

Reviewer #1 (Remarks to the Author):

I appreciate the great efforts that the authors have made in response to my questions and concerns. The manuscript has been substantially improved. I am for most part satisfied with their revisions. Still, I would like to invite the authors to address the two remaining concerns.

Ans.

Thank you very much for your encouraging comments. We have further revised our manuscript according to your suggestions.

Comment 1-2.

1. The authors concluded that the projection to the interpeduncular nucleus (IPN) underlies the aversive processing role of median raphe (MRN) 5-HT neurons. To support this conclusion, they used the inefficacy of the MRN projection to the medial habenula (MHb) as the negative example of non-IPN projections. However, previous tracing studies and Fig. 4 in the current study clearly show that MRN 5-HT neurons provide much denser output to the paraventricular thalamus (PVT) and the lateral habenula (LHb) than that to the MHb. In addition, numerous studies have well established that the PVT and the LHb participate in aversive processing. The conclusion would be much stronger if the authors can experimentally demonstrate that the serotonergic projections from the MRN to the PVT and the LHb are indeed not involved in aversive processing. At the very minimum, the authors should spell out the limitations of their current dataset and discuss the potential involvements of the MRN projections to other brain areas, including the PVT and LHb.

Ans.

We appreciate the reviewer for important comment. As the reviewer pointed out, we cannot rule out the possibility that MRN 5-HT neurons projecting to the PVT and LHb play a critical role in the processing of aversive information. Considering the previous reports showing the key role of these nuclei in the processing of aversive information, as the reviewer wrote, it is possible that MRN 5-HT neurons projecting to these nuclei respond to aversive stimuli but not to reward, as we found here.

Therefore, we added limitations of our study and further discussed potential critical roles played by MRN 5-HT neurons projecting to the PVT and LHb in the processing of aversive information according to the reviewer's suggestion as follows:

“Moreover, we cannot rule out the possibility that 5-HT^{MRN→PVT} and 5-HT^{MRN→LHb} play a critical role in the processing of aversive information in addition to 5-HT^{MRN→IPN}, while our data indicate these projections did not respond to reward. Furthermore, previous reports have revealed that LHb and PVT play an important role in the processing of aversive stimuli (Matsumoto and Hikosaka, Nature. 2007; Zhu et al., Nature. 2016 (PMID 26840481)). Therefore, further experiments to investigate the role of 5-HT^{MRN→PVT} and 5-HT^{MRN→LHb} in the processing of aversive information, such as morphine withdrawal, will be of high importance.”

Comment 1-3.

2. The authors now showed the change of Ca²⁺ signals in relation to the locomotor onset and offset. They conclude that MRN 5-HT neurons exhibit a weak but significant response to locomotor activity. However, they did not align the Ca²⁺ signals to change of locomotor speed, so it is unclear whether weak signals were due to small locomotor activity, and whether the signals will be substantially stronger if mice walked more rapidly. I suggest the authors to further discuss this possibility.

Ans.

We appreciate the reviewer for constructive comments. We agree with the reviewer's suggestion. According to the comment, we added a discussion on this possibility as follows:

“Additionally, we have to note that it is possible that the difference in the size of Ca²⁺ response to the initiation of locomotion and tail-pinch may be due to the difference in locomotor speed because we did not align the Ca²⁺ signals to change of locomotor speed.”

Responses to Reviewer #2 are as follows:

Reviewer #2 (Remarks to the Author):

The authors have satisfactorily addressed my comments. This is a very interesting and important study.

Ans.

We appreciate positive evaluation on our revised manuscript. We again thank the reviewer for their constructive comments on our original manuscript.